

# Machine learning techniques for imbalanced multiclass malware classification through adaptive feature selection

Binayak Panda[1], Sudhanshu Shekhar Bisoyi[2], Sidhanta Panigrahy[3] and Prithviraj Mohanty[2]

[1] Department of Computer Science and Engineering, Institute of Technical Education and Research, Siksha 'O' Anusandhan (Deemed to be) University, Bhubaneswar, Odisha, India
[2] Department of Computer Science and Information Technology, Institute of Technical Education and Research, Siksha 'O' Anusandhan (Deemed to be) University, Bhubaneswar, Odisha, India
[3] Haas School of Business, University of California, Berkeley, CA, United States

## ABSTRACT

Detecting polymorphic or metamorphic variants of known malware is an ever-growing challenge, just like detecting new malware. Artificial intelligence techniques are preferred over conventional signature-based malware detection as the number of malware variants proliferates. This article proposes an Adaptive Multiclass Malware Classification (AMMC) framework that trains base machine learning models with fewer computational resources to detect malware. Furthermore, this work proposes a novel adaptive feature selection (AFS) technique using the greedy strategy on term frequency and inverse document frequency (TF-IDF) feature weights to address the selection of influential features and ensure better performance metrics in imbalanced multiclass malware classification problems. To assess AMMC's efficacy using AFS, three open imbalanced multiclass malware datasets (VirusShare with eight classes, VirusSample with six classes, and MAL-API-2019 with eight classes) on Windows API sequence features were used. Experimental results demonstrate the effectiveness of AMMC with AFS, achieving state-of-the-art performance on VirusShare, VirusSample, and MAL-API-2019 with a macro F1-score of 0.92, 0.94, and 0.84 and macro area under the curve (AUC) of 0.99, 0.99, and 0.98, respectively. The performance measurements obtained with AMMC for all datasets were highly promising.

# INTRODUCTION

The internet has become increasingly prevalent in daily life due to advancements in information and communication technology. People of all ages rely on various applications operating on various computing devices to meet their daily needs, such as transportation, health, banking, and retail. The increasing use of apps poses security risks for devices and applications. Malicious software, also known as malware, poses the greatest

Corresponding author
Sudhanshu Shekhar Bisoyi,
sudhanshu.bisoyi@gmail.com

threat to the cyberworld. Malware programs evolve as technology advances, threatening our security and privacy. Malware is increasingly being used to carry out a variety of destructive operations on victims' devices. Malware causes anomalies in the functioning of a computer system, raising security concerns. As the number of malware variants proliferates, conventional signature-based malware detection has become less effective and time-consuming. Artificial intelligence techniques, such as machine learning and deep learning, are becoming increasingly popular for studying and analyzing malware behavior to detect it more efficiently. The malicious payload is delivered to the affected device *via* different routes, including email attachments, advertisements, potentially unwanted apps (PUAs), and free utility software. Malcoders typically obfuscate malware families such as Trojan, Backdoor, Spyware, and Worm, posing serious threats to system vulnerabilities (*Yan et al., 2023*). According to the Quick Heal annual report FY-2022–23, a total of 163.81 million of malware from the families Trojan, Infector, PUA, Worm, Cryptojacking, Ransomware, and Adware were detected during 2021 and 2022. Figure 1 shows the family-wise malware detection statistics for 2021 and 2022. Figure 2 shows the top five malware from July to September 2022. Furthermore, the report shows a substantial number of instances in each class of malware with an exceptional increase in instances of Trojan and Cryptojacking (*QuickHeal, 2022*).

Protecting devices from malware is possible by using the time-consuming and expensive statistical methods that are really complex in nature. Malware detection is a computationally hard problem, but the use of statistical methods can restrict undetected spreading (*Cohen, 1987*). A lot of efforts are being made to limit the spread of malware. In this context, classifying malware against benign and malware against malware is an ongoing need. There are two basic approaches, named static and dynamic, to analyze malware. Static analysis is about studying some key features, like opcode sequences, printable strings, *etc.*, of the malware without executing it. It requires some reverse engineering techniques that can disassemble them. However, there may be challenges with packed malware when performing static analysis on it. On the contrary, dynamic analysis is about collecting execution time features like system call graph, application programming interface (API) sequence, registry file contents, *etc.* by allowing the malware to execute in a controlled environment. However, the malware may not exhibit its original execution pattern when executed under a controlled environment (*Damodaran et al., 2017*; *Shibahara et al., 2016*). Many researchers have applied machine learning techniques to mitigate the challenges of analyzing metamorphic and polymorphic malware variants (*Wong & Stamp, 2006*). Developing an effective malware detection system is tough, especially when dealing with newer threats. Advancements in virus evasion techniques have led to significant consequences. Advanced malware detection technologies, such as machine learning and deep learning, overcome the limitations of traditional methods (*Gopinath & Sethuraman, 2023*). The authors examine current malware detection algorithms for Android OS, i-OS, IoT systems, Windows OS, advanced persistent threats (APTs), and ransomware. Many dynamic analysis algorithms referred API call sequences to be a crucial behavioral characteristic that distinguishes malicious programs from benign

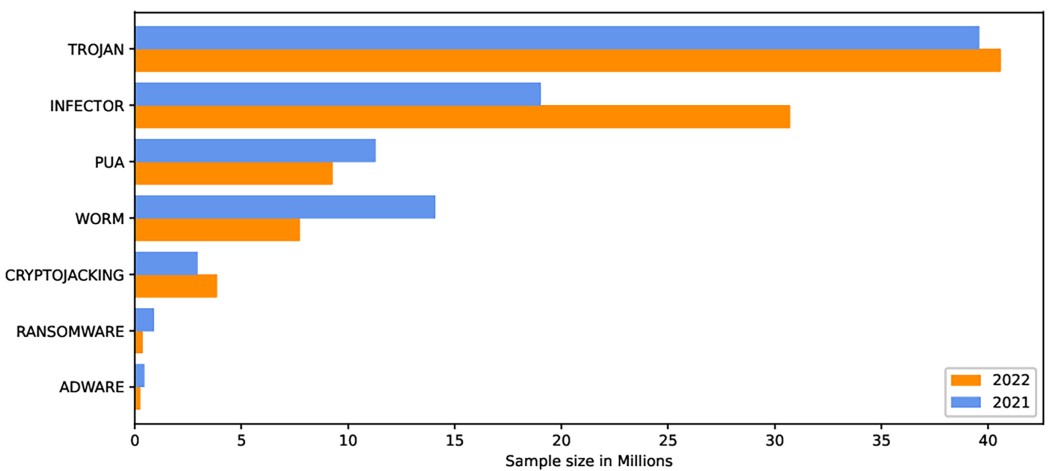

**Figure 1 Family wise malware detection statistics Quick Heal annual report FY-2022–23.**

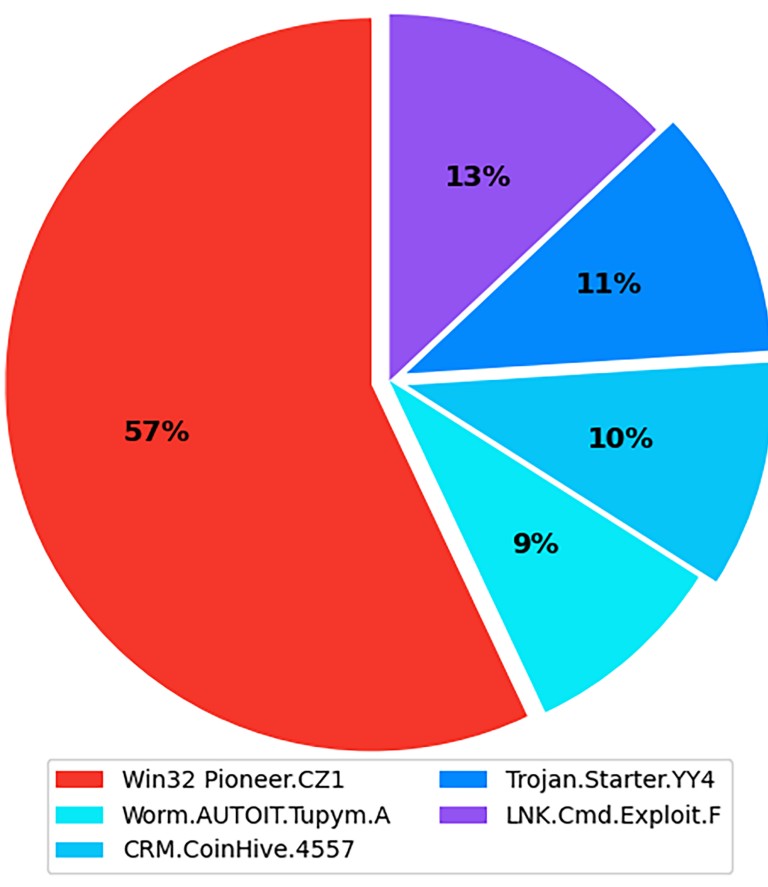

**Figure 2 Top 5 malware from July to September 2022.**

ones utilizing machine learning and deep learning approaches (*Gaber, Ahmed & Janicke, 2024*). Windows as an operating system provides APIs for a variety of operations pertaining to files, networks, I/O, and processes. Any program, including malicious

software, calls APIs sequentially to do the required tasks. An essential feature set for analyzing program activity is the API sequence. In order to avoid detection, malware in particular is designed to call APIs frequently and sporadically. The analysis of benchmark multiclass malware datasets such as VirusSample, VirusShare (*Düzgün et al., 2021*), and MAL-API-2019 (*Ferhat Ozgur et al., 2020*) reveals that the length of the API sequences is either excessively long due to repeated calls or insufficient. The dataset is quite challenging because of the wide variation in API sequence length. The goal of this work is to improve performance metrics with regard to multiclass malware datasets that are imbalanced. The objectives of this work are:

1. Developing a feature selection strategy to address the issue of feature imbalance related to the diversity of API sequence lengths in malware traces.
2. Developing a unique adaptive feature selector (AFS) that employs a greedy method to improve the classification metrics for imbalanced multiclass malware classification problems.
3. Designing a framework called Adaptive Multiclass Malware Classifier (AMMC) to train base machine learning models that use fewer resources, ensuring better classification metrics.

"Related Work" will explore related works. "Methodology" elaborates the working principles of AFS and AMMC. "Experimental Evaluation" discusses the dataset's description and the experimental setup with results. The concluding observation is highlighted in the 'Conclusion'.

## RELATED WORK

Recent malware research has revealed that machine learning and deep learning are often used for malware investigation and categorization. API sequences for malware categorization aim to enhance accuracy and efficiency through advanced deep learning and machine learning methods.

*Ye et al. (2008)* gathered API call data from PE files and created a verifiable and comprehensible feature set. They developed an intelligent malware detection system based on objective-oriented association (OOA) classification. These features were utilized to determine whether the PE file is malicious or not. They have not predicted different types of malware. *Vinod et al. (2010)* used dynamic analysis of four metamorphic classes of malware to create signature-based malware fingerprints to trace their API sequences. Their experiment used the API sequences of 80 viruses from four families and 20 harmful apps. They used the Chi-square test to determine how closely a piece of malware matched a given class. This method achieved 75%, 75%, 80%, and 80% accuracy rates for the families NGVCK, IL-SMG, G2 and MPCGEN. They stated that increasing the number of samples will enhance precision. The signature matching approach is susceptible to recently identified malware samples. *Ding et al. (2013)* suggested an API based association mining strategy that eliminated infrequent items from the API sequences. They selected association rules with good classification abilities and implemented them to increase

detection accuracy. For classifying malware into 11 families, *Kong & Yan (2013)* combined several malware attributes like API calls, registers, opcodes, *etc*. They developed an effective system that was capable of recognizing previously unidentified samples using discriminant distance metric learning, pairwise graph matching, and ensembled classification (*Kong & Yan, 2013*).

*Mehra, Jain & Uppal (2015)* obtained control flow and API call graphs from 600 malware and 150 benign samples. They suggested utilizing Gourmand Feature Selection to extract desired attributes from API call graphs. The WEKA classification tool yielded 89%, 91.08%, 92.24%, 94.56%, and 99.1% accuracy rates for KNN, SMO, VP, NB, and J-48 classifiers, respectively. They used portable executables instead of parsing API sequences generated in runtime. A supervised approach, like Random Forests technique, was used by *Pirscoveanu et al. (2015)* for the classification of malware. The Cuckoo Sandbox application was used to gather a total of 42,000 malware behaviours. Windows API calls were categorised using DNS requests, files that were accessed, mutexes, and registry key information. Additionally, the labels recognised by the Avast applications outperform the outcomes offered by the VirusTotal service for the classes of harmful software. System call sequences were examined by *Kolosnjaji et al. (2016)* to classify malware. Convolutional and recurrent network layers were used to extract the best characteristics. Using this hybrid neural network architecture, they were able to average precision of 85.6% and recall of 89.4%. *Zhang et al. (2016)* proposed a simple ensemble learning algorithm for the classification of malware, utilizing data from the Microsoft malware classification challenge on Kaggle. Malware samples from the unbalanced training dataset were accurately classified into the proper family.

According to recent studies on malware analysis, deep neural networks and supervised and unsupervised machine learning techniques are frequently employed to detect malicious activities with higher accuracy, efficiency, and a lower false positive rate. The primary components of machine learning-based malware detection techniques are automatic detection and feature extraction (*Ye et al., 2017*). Due to the scalability, speed, and flexibility, machine learning algorithms like logistic regression (LR), support vector machine (SVM), Random Forest (RF), K-nearest neighbor (KNN), frequent pattern mining (FPM), *etc*. are frequently used to discover and categorize unidentified samples for malware family. *Han et al. (2019)* applied the TF-IDF technique to 807 benign and 3,027 malicious (including packed and unpacked variants) samples. The MalDAE framework uses machine learning techniques such as KNN, decision tree (DT), extreme gradient boosting (XGBoost), and RF to identify and categorize malware based on static, dynamic, and fused API sequences. Dynamic API sequences outperformed static API sequences with accuracy rates of 74.74%, 79.65%, 83.15%, and 85.96%, respectively. However, with the fused API sequence, accuracy improved to 85.26%, 88.42%, 93.33%, and 94.39%, respectively. Given the difficulty posed by the enormous number of malware kinds, deep learning techniques were suggested to boost efficiency.

*Huda et al. (2016)* suggested an automated malware detection method to calculate statistical information on API calls performed by malicious and benign programs. They

developed two novel hybrid API feature selection methods based on hybrid SVM encapsulation heuristic method and maximum associated or minimal redundancy filtering heuristics approach to determine the best API calls to detect malware. Hidden Markov models (HMMs) were utilized by *Damodaran et al. (2017)* on their own multiclass malware dataset, and it has been concluded that the dynamic approach based on API calls was quite successful. *Ucci, Aniello & Baldoni (2019)* stated that a malicious program can be obfuscated as a new program that is mistakenly labelled as benign while retaining the original behaviour and its consequences. The detection process for this new software might be readily avoided. *Or-Meir et al. (2019)* have stated that code obfuscation technique is less effective on dynamic analysis as compared to static analysis. The heuristic-based malware detection algorithms often transform a malware sample into a basic form, such as text, picture, or signal, before extracting essential aspects for further analysis (*Hashemi & Hamzeh, 2019*; *Mohammadi et al., 2019*; *Fu et al., 2018*). The time needed to modify the sample representation as a system call, opcode sequence, *etc.* and the time required for feature extraction resulted in substandard performance on both the training and testing phases of the algorithms, even though such methods might achieve decent accuracy. Several researchers have tried to use natural language processing (NLP) techniques on malware static properties such as Opcode sequences and file header sections to identify malicious programs. Using feature extraction techniques like n-grams, term frequency and inverse document frequency (TF-IDF), *etc.*, they discovered low detection accuracy— below 87%. To perform static analysis on the visual malware pictures used in their recurrent neural network, Minhash, Visualization, and convolutional neural network (RMVC) approach, *Sun & Qian (2021)* used recurrent neural network (RNN) and convolutional neural network (CNN). They found accuracy better than 92%, even with a tiny training dataset. Experimental results show that the classification accuracy of malware classes in the dataset with noise factor is 0.96, higher than the accuracy of 0.83 in a dataset without noise factor. They suggested that a potential avenue for future research was to examine the effectiveness of their approach in dynamic analysis. Malware was categorized using images produced from malware binaries by *Hammad et al. (2022)*. KNN, SVM, and the elaboration likelihood model (ELM) are trained and tested using the Malimg dataset. Features are extracted using GoogleNet, a deep learning model, and Tamuar, a texture feature that correlates with human visual perception. They found that ELM performed better than any other model. They have recommended using data augmentation, which could enhance classification outcomes.

System call sequences from 3,536 benign and 3,567 malicious applications for Android-OS were employed by *Xiao et al. (2019)* on the long short-term memory (LSTM) model. They discovered a low FPR of 9.3% and a high recall of 96.6%. Two LSTM models make up their classifier: one is used to train the malware, while the other is used to train the trusted application. New sequences were classified using the similarity score that was derived from the two models. *Mathew & Ajay Kumara (2020)* have employed TF-IDF embeddings with N-grams for feature extraction and selection. Using API call sequences, the proposed LSTM model is used to classify applications as benign or malicious. The authors obtained a

92% accuracy score on unidentified test API call sequences. The feature that represents hidden malicious behavior plays a vital role in malware detection. The insignificant feature leads to poor classification performance. To increase classification performance, eliminate insignificant and noisy features from the dataset. *Bhat, Behal & Dutta (2023)* suggested a very stable and reliable malware binary classification model based on the ensemble technique. To improve intrusion detection systems' accuracy, *Taheri, Ahmadzadeh & Kharazmi (2015)* employed the Cuttlefish algorithm for feature reduction. Four distinct datasets, KDD5000, KDD10000, KDD100000, and KDD500000, were used for selecting various feature numbers, such as 3, 5, 10, 13, and 41. They reported promising results using the dataset and an artificial neural network as the evaluation function. *Taheri et al. (2024)* have investigated two adversarial techniques, Data Poisoning with Noise Injection (DP-NI) and Gradient-based Data Poisoning (GDP), to assess the vulnerability of deep learning-based Android malware classifiers. They have suggested a novel defense technique called Differential Privacy-Based Noise Clipping (DP-NC) to make Android malware classifiers more resilient to these adversarial attacks. By employing adversarial training techniques and deep neural networks, DP-NC demonstrates remarkable efficacy in mitigating the impact of GDP and DP-NI attacks. *Panda & Tripathy (2020)* have created a host-based anomaly detection system by using the TF-IDF word embedding approach on the DLL sequence of all in-memory processes to identify abnormal processes. *Panda, Bisoyi & Panigrahy (2023)* have trained multiple 1D-CNN models using the one-*vs.*-rest strategy and generated API embeddings using the Word2Vec word embedding technique. They proposed ModifiedSoftVoting to integrate classification capabilities in multiclass malware classification challenges to improve classification metrics.

*Ferhat Ozgur et al. (2020)* published a dynamic data set named MAL-API-2019 using the Cuckoo Sandbox, comprising 7,107 API sequences for eight different forms of malware. The data set has also been trained to perform multiclass classification using single-layer LSTM, two-layer LSTM, SVM, KNN, RF, and DT. Compared to all other models, the single-layer LSTM achieved macro F1-score of 0.47 as the maximum. *Li & Zheng (2021)* used LSTM and gated recurrent unit (GRU) models on the benchmark dataset MAL-API-2019 to classify malware classes using long-sequence API calls. They have achieved a precision of 0.56 in both approaches and a recall of 0.58 and 0.59, respectively, in LSTM and GRU. The broad applicability of LSTM and the best architecture for malware classification were examined by *Avci, Tekinerdogan & Catal (2023)*. Using the MAL-API-2019 dataset, they evaluated and compared the performance of several LSTM architectures, such as CNN-LSTM, stacked-LSTM, bi-directional LSTM, and vanilla-LSTM. In order to produce comprehensible results and classification metrics, *Galli et al. (2024)* integrated the eXplainable artificial intelligence (XAI) technique with an AI-based malware detection process. They evaluated various deep learning models on the dataset MAL-API-2019, BiLSTM and achieved the best experimental classification results, with accuracy, precision, recall, and F1-scores of 52.88%, 54.89%, 53.99%, and 54.31%, respectively. CAFTrans, a framework that processes API sequences using CNN and LSTM networks, was proposed by *Qian & Cong (2024)*. The MAL-API-2019 dataset was used to

**Table 1  State-of-the-art approaches for multiclass malware classification with API sequences.**

| Author | Description | Method |
|---|---|---|
| *Ye et al. (2008)* | Developed an intelligent malware detection system based on Objective-Oriented Association (OOA) classification. | OOA |
| *Mehra, Jain & Uppal (2015)* | They suggested utilizing Gourmand Feature Selection to extract desired attributes from API call graphs. | KNN |
| *Huda et al. (2016)* | Proposed an automated malware detection method to calculate statistical information on API calls performed by malicious and benign programs. | SVM |
| *Pirscoveanu et al. (2015)* | Classify the windows API calls using DNS requests, files that were accessed, mutexes, and registry key information. | Random Forest |
| *Manavi & Hamzeh (2022)* | The headers of executable files, more significantly portable executable files, are used to detect ransomware. A graph is then made using the headers of executable files, and the "Power Iteration" approach is used to map the graph in an eigenspace. | |
| *Kolosnjaji et al. (2016)* | Examine the System call sequences to classify the malware using hybrid neural network. | CNN + RNN |
| *Sun & Qian (2021)* | The techniques of recurrent neural networks (RNN) and convolutional neural networks (CNN) are combined with the static analysis of malicious code to generate malware feature images. | |
| *Kong & Yan (2013)* | They developed an effective system that was capable of recognizing previously unidentified samples. | Ensemble |
| *Zhang et al. (2016)* | Classify the malware samples from the unbalanced training dataset accurately. | |
| *Bhat, Behal & Dutta (2023)* | Suggested a very stable and reliable malware binary classification model. | |
| *Ferhat Ozgur et al. (2020)* | Using sequential data taken from Windows operating system API calls, the LSTM model is utilized to categorize malware into eight families. | |
| *Xiao et al. (2019)* | System call sequences from 3,536 benign and 3,567 malicious applications for Android-OS were employed on LSTM for classification. | LSTM |
| *Mathew & Ajay Kumara (2020)* | Employed TF-IDF embeddings with N-grams for feature extraction and selection. | |
| *Li & Zheng (2021)* | Employed LSTM and GRU (Gated Recurrent Unit) models on the benchmark dataset MAL-API-2019 to classify malware classes using long-sequence API calls. | |
| *Avci, Tekinerdogan & Catal (2023)* | Examined the broad applicability of LSTM and the best architecture for malware classification. | |
| *Galli et al. (2024)* | Proposed a framework called as CAFTrans that processes API sequences using CNN and LSTM networks and asserted more accurate classification of malware. | |
| *Galli et al. (2024)* | Integrated the eXplainable Artificial Intelligence (XAI) technique with an AI-based malware detection process. | BiLSTM |
| *Demirkiran et al. (2022)* | Classified malware into multiple classes using the benchmark datasets VirusShare, VirusSamle, and MAL-API-2019. | RTF |
| *Miao et al. (2024)* | Proposed a lightweight malware detection model based on knowledge distillation that exhibited a superior classification performance. | DistillMal and BERT |

test the framework, and the results showed an F1-score of 0.65 and an area under the curve (AUC) of 0.89. They asserted that by identifying malware in their families more accurately than in others on the same dataset, CAFTrans improves accuracy.

*Düzgün et al. (2021)* published two datasets VirusShare and VirusSample. These are created using MD5 malware hashes, with 14,616 samples taken from VirusShare and 9,795 from VirusSample sites, respectively. The malware samples' API calls are retrieved using the Python module PEfile, which extracts API calls from a program's Portable Executable

(PE) file header. These static analysis techniques result in static API call sequences. They have used the balanced version of these two datasets on several machine learning and deep learning models. On VirusShare with SVC, they achieved the highest macro F1-score of 0.76, whereas on VirusSample, the highest macro F1 the CANINE model is 0.91. *Demirkiran et al. (2022)* classified malware into multiple classes using the benchmark datasets VirusShare, VirusSamle, and MAL-API-2019. Their suggested RTF model outperforms the Transformer, CANINE-S, and BERT deep learning models. With the RTF model, they have achieved the highest macro F1-scores on the benchmark dataset, which are 0.72, 0.80, and 0.61, respectively. *Miao et al. (2024)* proposed a lightweight malware detection model based on knowledge distillation (DistillMal) that exhibited superior classification performance in several evaluation metrics when applied to VirusShare and VirusSample datasets. On the VirusSample dataset, DistillMal's experimental performance metrics showed an accuracy of 0.94, a macro precision of 0.50, a macro recall of 0.73, and a macro F1-score of 0.60. In contrast, VirusShare's accuracy was 0.89, accompanied by a macro F1-score of 0.69, macro recall of 0.63, and macro precision of 0.88.

The challenge of the imbalanced multiclass malware classification problem is addressed using various strategies, with the API sequence being the most crucial feature sequence. Table 1 summarises works by several other researchers considering API sequence as a critical feature. This study solves the issue of feature imbalance concerning diversity in API sequence length of the multiclass malware classification problem using the unique feature selection technique AFS. As machine learning models use fewer resources to train than deep learning models, the proposed AMMC framework trains base machine learning models using Skip-gram API embeddings to ensure superior classification reports with fewer resources. Using base machine learning models instead of deep learning or other complicated models reduces the higher resource requirements for malware detection.

## METHODOLOGY

The AMMC workflow is depicted in Fig. 3 as a sequence of stages. This framework takes a labeled imbalanced *Multiclass Malware API Sequence Dataset* as input. It ensures a better classification result through a novel AFS process that works iteratively. In the *Nonconventional API Name Removal* stage, each API sequence in the dataset is scanned to drop the nonconventional API names. A nonconventional API name may contain symbols beyond the allowed symbols $[A..Z]$, $[a..z]$, $[0..9]$, and $[\_]$. Such API names are found in the case of static API sequence datasets.

API call sequences, in both static and dynamic datasets, tend to be lengthy due to repeated occurrences of identical API calls. A notable characteristic of malware behaviour is the repetition of specific API calls within these sequences to evade detection mechanisms. Each API call corresponds to a machine-level task, and analyzing the sequence of unique calls provides valuable insights into malware activity. The proposed AMMC framework introduces a *Duplicate API Removal* step to address redundancy in API sequences before AFS. This step ensures the preservation of the first occurrence of each distinct API call while eliminating subsequent duplicates. By focusing on unique API

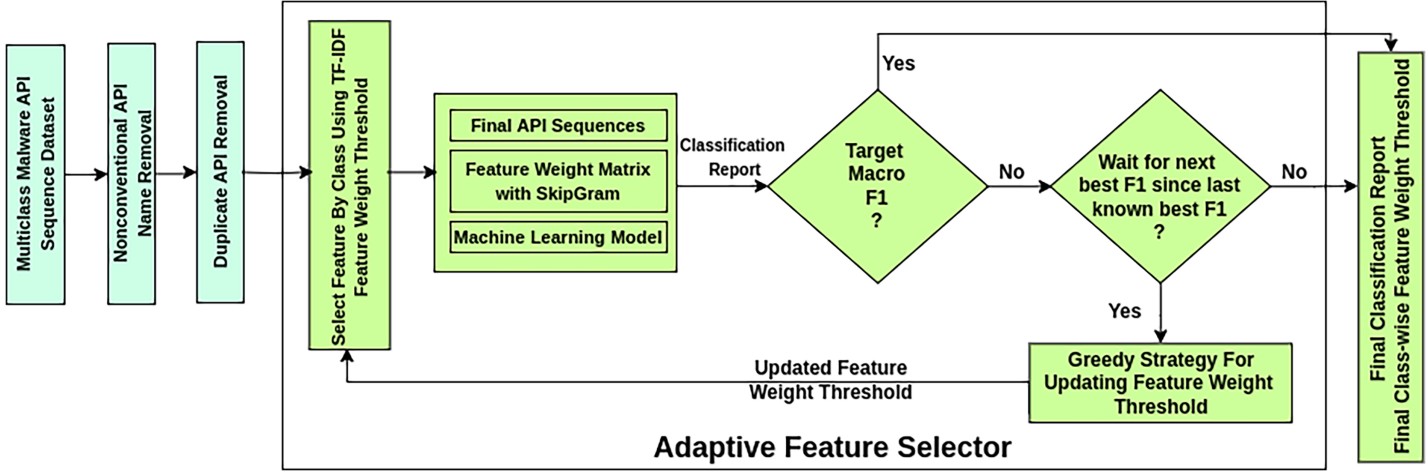

**Figure 3 Workflow of adaptive multiclass malware classifier.**

calls, the sequence reflects the underlying machine-level tasks performed by the malware more accurately. For instance, consider an encoded API sequence: *[2, 2, 2, 5, 5, 2, 2, 5, 1, 1, 2, 2, 5, 2, 5, 1, 1, 4, 4, 2, 2, 2, 5, 1, 1, 1, 1, 4, 4, 4, 4, 7, 7, 1, 1, 4, 4, 2, 2, 5, 2, 2, 5, 1, 1, 1, 4, 4, 4, 2, 2, 2, 5, 5, 2, 2, 5, 5, 2, 2, 5, 1, 1, 1, 4, 4, 4, 7, 7, 1, 1, 4, 1, 1, 4]*. After applying duplicate API removal, the processed sequence becomes: *[2, 5, 1, 4, 7]*. This step reduces the sequence size and ensures that critical, non-redundant information about the malware's operations is retained, enhancing the efficiency of subsequent analysis stages.

The AFS stage in the framework works iteratively over three substages. The substage Select Feature by Class selects influential APIs in an API sequence considering the TF-IDF feature weight threshold. During influential feature selection, it mitigates the feature sequence imbalance problem concerning API sequence length. At the same time, it preserves the class-wise sample size. The substage *Machine Learning Model* trains and tests specific classifiers on the finalized API sequences. Considering the classification report of the model, the substage *Greedy Strategy for Updating Feature Weight Threshold* updates the feature weights for the following selection of influential features by *Select Feature by Class*. A new goal is set to obtain better classification results once a desired macro F1-score is reached. The best-found classification report of all iterations is produced if the desired macro F1 is not reached after a predetermined number of iterations.

The AFS adapts changes in feature selection criteria to achieve better performance. The functioning of the AFS is detailed in Algorithm 1. It necessitates three inputs: $F_1^t$, $W_{cnt}$, and $D_{api}$. $F_1^t$ is to be provided as a target for the macro F1-score and is expected to be achieved by AFS. In the event that the maximum macro F1 ($F_1^{max}$) achieved does not reach the target, the $W_{cnt}$ constant is supplied to end the search for $F_1^t$ instead of going into an infinite loop. $D_{api}$ is a benchmark multiclass malware dataset with $n$ classes. Listed are the data structures in statement 1 through statement 4 used in algorithm AFS to search for $F_1^t$. $F_1^{max}$ is initialized to 0 and used to keep track of the maximal macro F1-score achieved over the iterations against the set $F_1^t$. $Wait_{cnt}$ is initialized to $W_{cnt}$ and is used to restrict the

---

**Algorithm 1   AdaptiveFeatureSelector.**

**Require:** $F_1^t$, $W_{cnt}$, $D_{api}$

**Ensure:** $cr$, $CFWT$, $F_1^{max}$

1:  $F_1^{max} = 0$           ▷ To track maximal macro F1 *i.e.*, $F_1^{max}$ score achieved so far

2:  $Wait_{cnt} = W_{cnt}$           ▷ Count to restrict from entering to an infinite loop

3:  $CFWT[1 .. n] = \{0.0\}$         ▷ To track class wise feature weight threshold

4:  $F_1[1 .. n + 1] = \{0.0\}$         ▷ To track class wise and macro F1-scores

5:  **while** True **do**

6:       $D_{api}^F = SelectFeatureByClass(D_{api}, CFWT)$

7:       $cr = TrainTestModel(D_{api}^F)$

8:       $F_1[1 .. n + 1] = CollectF_1Score(cr)$

9:       **if** $F_1[n + 1] >= F_1^{max}$ **then**        ▷ Check current iterations macro F1 is better than $F_1^{max}$ or not

10:          $F_1^{max} = F_1[n + 1]$

11:          $CFWT_{sel} = CFWT$

12:          $cr_{sel} = cr$

13:          $Wait_{cnt} = W_{cnt}$

14:      **else**

15:          $Wait_{cnt} = Wait_{cnt} - 1$

16:      **end if**

17:      **if** $Wait_{cnt} == -1$ **then**        ▷ $Wait_{cnt}$ exhausted, $F_1^t$ not achieved and return maximal macro F1

18:          Return $(cr_{sel}, CFWT_{sel}, F_1^{max})$

19:      **end if**

20:      **if** $F_1[n + 1] <= F_1^t$ **then**

21:          **for** $i = 1 \rightarrow n$ **do**

22:              **if** $F_1[i] <= F_1[n + 1]$ **then**

23:                  $CFWT[i] = CFWT[i] + 0.01$

24:              **end if**

25:          **end for**

26:      **else**

27:          $F_1^t = F_1^t + 0.05$        ▷ $F_1^t$ achieved and adjust it to seek superior macro F1

28:      **end if**

29: **end while**

---

search process entering into an infinite loop while the targeted $F_1^t$ is not becoming realizable. $CFWT[\ ]$ is an array of size n to define the classwise feature weight threshold, and all elements are initialized to 0. CFWT gets updated after each iteration to tighten classwise feature section criteria. $F_1[\ ]$ is an array of size $n + 1$ where the first $n$ elements represent F1-scores of $n$ classes and the last element represents macro F1-score of the

---

**Algorithm 2** SelectFeatureByClass.

---

**Require:** $D_{api}$, $CFWT$

**Ensure:** $D_{api}^F$

1:   $D_{api}^F = \phi$

2:   **for** $C_{id} = 1 \rightarrow n$ **do**

3:        $d_{c_{id}}$ = select $API_{seq}$ from $D_{api}$ by $C_{id}$

4:        $W_{C_{id}}[\,][\,]$ = $TFIDFWeightEvaluator(d_{c_{id}})$

5:        $d_{c_{id}} = SelectFeature(d_{c_{id}}, W_{C_{id}}, CFWT[C_{id}])$

6:        $D_{api}^F = D_{api}^F \cup d_{c_{id}}$

7:   **end for**

8:   return $D_{api}^F$

---

classifier. The scores in F1[ ] are used to keep track of improvements while searching for the $F_1^t$ of the classifier.

AFS iterates around statement 5 to statement 29 in the while loop to find the $F_1^{max}$ score. By choosing the features from dataset $D_{api}$ that meet the corresponding weights for each class in CFWT, *SelectFeatureByClass* generates the final dataset $D_{api}^F$. *TrainTestModel* builds the classifier and produces the classification report ($cr$) using the dataset $D_{api}^F$. *CollectF$_1$Score* creates the F1-score array. Furthermore, the classifier's macro F1 is compared against $F_1^{max}$ to record any improvement. The classifier's macro F1-score is compared to the $F_1^t$ in each iteration to determine additional weight adjustments in *CFWT*. When a class's F1-score is less than the classifier's macro F1-score, the feature weight for the classes is incremented by *0.01*. $F_1^t$ is incremented by *0.05* to look for a better macro F1-score if the $F_1^{max}$ score reaches the $F_1^t$. Lastly, the $F_1^{max}$ score with the associated $cr$ calculated in addition to the *CFWT* weights is returned if the $F_1^t$ is not realized for the $W_{cnt}$ number of iterations after achieving a $F_1^{max}$ score.

In order to solve the feature imbalance issue brought on by the variety of API sequence lengths in the $D_{api}$, which poses a challenge to the classifier's effectiveness, the algorithm AFS calls the subroutine *SelectFeatureByClass* in statement 6. This subroutine chooses significant APIs from API sequences as outlined in Algorithm 2. The statements of *SelectFeatureByClass* are explained in detail below.

- Statement 3 selects $API_{seq}$ from $D_{api}$, where each $d_{c_{id}}$ will hold the API sequences of the $D_{api}$ whose class is $C_{id}$. It divides the $D_{api}$ into several disjoint datasets $d_{c_1}, d_{c_2}, \ldots, d_{c_{id}}$ where $d_{c_{id}}$ is the API data for class $C_{id}$.

- Statement 4 uses a TF-IDF weighting schema to select APIs in $API_{Seq}$ that qualify the mentioned weights in CFWT by $C_{id}$. It calculates a weight matrix (*i.e.*, $W_{C_{id}}[\,][\,]$) for every $d_{c_{id}}$ in order to address the data imbalance problem associated with the variety of API sequence lengths. The TF-IDF weighting schema is applied to each $d_{c_{id}}$ to decide the weights for each distinct API name concerning the API sequence in $d_{c_{id}}$. TF-IDF is a

---

**Algorithm 3  TrainTestModel.**

**Require:** $D_{api}^F$

**Ensure:** $cr$                ▷ $cr$: classification report

1:   $f_{wt}[\,][\,] = Word2vec.SkipGram(D_{api}^F)$        ▷ $f_{wt}$: Feature weight matrix

2:   $D_{tr}, D_{ts} = TrainTestSplit(D_{api}^F, 0.8)$        ▷ $D_{tr}$: Train set and $D_{ts}$: Test set

3:   $Model.train(D_{tr}, f_{wt})$

4:   $cr = Model.predict(D_{ts}, f_{wt})$

5:   return $cr$

---

statistical measure that ensures the importance of a word (*i.e.*, a distinct API) to a document (*i.e.*, $API_{seq}$) in the collection of documents (*i.e.*, $d_{c_{id}}$). Considering TF-IDF over raw frequencies of occurrences of APIs is to scale down the impact of very frequently occurring APIs in an $API_{seq}$ in the collection of API sequences, which is empirically less informative than the APIs of less frequency. The weight of $i$th API to $j$th API sequence is denoted as $W_{ij}$ in the $W_{C_{id}}[\,][\,]$ and defined as given in Eq. (1).

$$W_{ij} = TF_{ij} \; X \; IDF_i \tag{1}$$

$TF_{ij}$ in Eq. (1) is the term frequency for $i$th API with respect to the $j$th API sequence and defined as given in Eq. (2).

$$TF_{ij} = \frac{Number \; of \; times \; \mathbf{i}^{th} \; API \; appears \; in \; \mathbf{j}^{th} \; API \; Sequence}{Total \; number \; of \; APIs \; in \; \mathbf{j}^{th} \; API \; sequence} \tag{2}$$

$IDF_i$ in Eq. (1) quantifies $i^{th}$ API's infrequency throughout $d_{c_{id}}$. It is determined by taking the logarithm of the ratio between the total number API sequences in $d_{c_{id}}$ and the number of API sequences that contain the $i^{th}$ API as given in Eq. (3). Penalizing APIs that appear in every API sequence is the aim.

$$IDF_i = \log\left(\frac{Total \; number \; API \; sequences \; in \; \mathbf{d}_{c_{id}}}{Number \; of \; API \; sequences \; that \; contain \; the \; \mathbf{i}^{th} \; API}\right) \tag{3}$$

- The finalized $d_{c_{id}}$ is found by iterating over each API sequence and retaining those APIs whose weight as per the $W_{C_{id}}[\,][\,]$ qualifies the weight threshold stated in $CFWT[C_{id}]$. This is done using statement 5. If none of the APIs meet the threshold, a preventive measure is also taken to return the sequence to its previous state. This prevention step keeps shorter API sequences from being excluded and preserves the sample size of $d_{c_{id}}$.

- In statement 6, the final dataset $D_{api}^F$ is constructed by combining each of the classwise finalized datasets $d_{c_{id}}$. This final dataset ($D_{api}^F$) is returned to train and validate a classifier.

AFS employs the *TrainTestModel* as outlined in Algorithm 3 to ensure several performance metrics by training and evaluating a classifier on $D_{api}^F$. It returns the

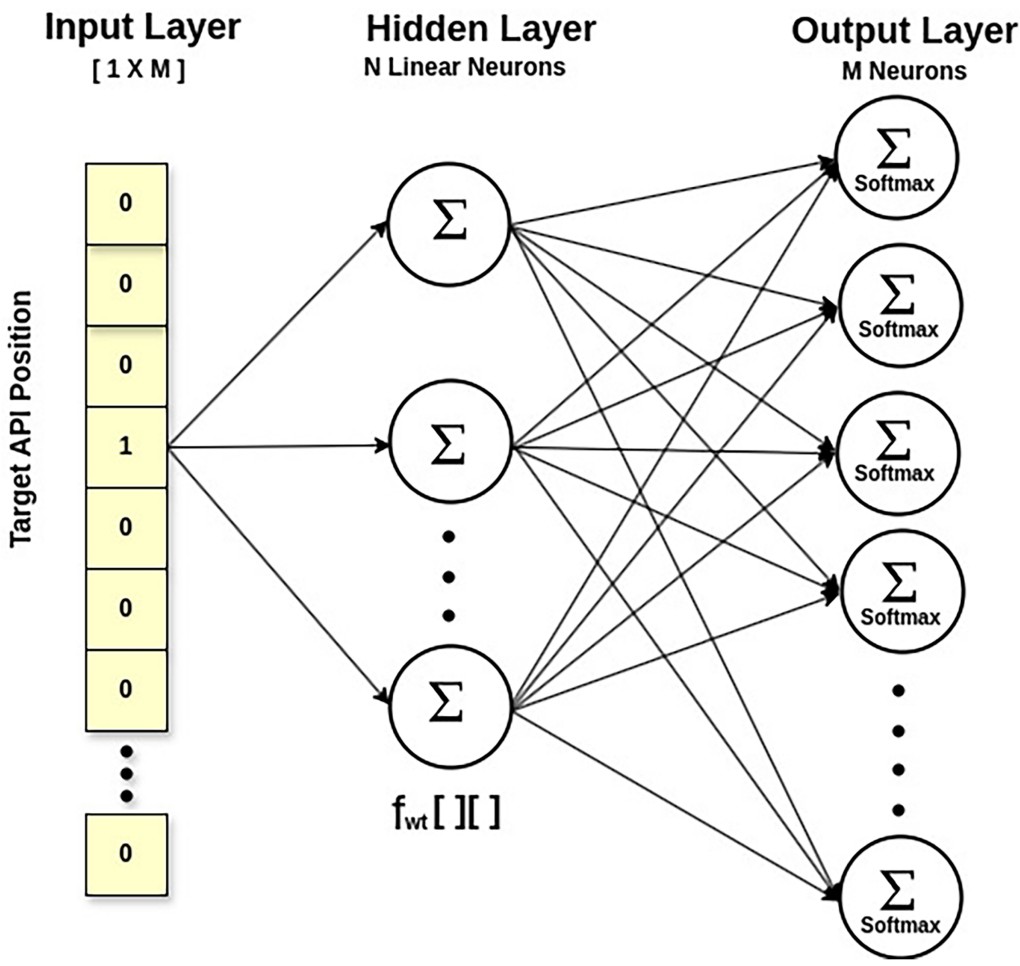

**Figure 4 SkipGram model for API feature weight matrix.**

classification report ($cr$) to AFS for further improvements in $CFWT$ to meet the $F_1^t$. The steps of *TrainTestModel* are explained below in detail.

- Statement 1 calculates $f_{wt}[\ ][\ ]$ the feature matrix of API embeddings *via* the *Skip-Gram* approach of *Word2Vec*, a neural network model (*Mikolov et al., 2013*). Technically, it calculates the likelihood of an API being a context API for a specified target API. This way, it also explores the multicollinearity between APIs, taking their semantic relationship. The output probabilities generated by the network will tell us how likely it is to find each vocabulary API near our input API. Figure 4 displays a shallow network with an input layer, a single hidden layer, and an output layer. The fascinating thing is that we do not employ the trained neural network. Instead, the goal is to learn the hidden layer's weights while predicting the surrounding APIs. These weights represent API embeddings. The input is a one-hot encoded vector corresponding to the API vocabulary size in $D_{api}^F$, with a specific API being the target API. The output layer is made up of neurons equal to the API vocabulary size, and the *Softmax* activation function is used to

calculate the appropriate probability for each API in the API vocabulary. The hidden layer is linear, and its optimal weights are used to generate the learned API embeddings. For example, API embedding of a target API will be a vector of dimension $[1\ X\ N]$, obtained by multiplying $[1\ X\ M]$ matrix (the one-hot vector of the target API) with $[M\ X\ N]$ matrix (the feature weight matrix $f_{wt}[\ ][\ ]$ at the hidden layer), where 'M' is the API vocabulary size and 'N' is the number of neurons in the hidden layer.

- In statements 2 and 3, the usual 80:20 split ratio for training and testing data is considered during training and evaluating a model. While training, feature weight vectors are normalized, and hyperparameters are tuned with the grid search, emphasizing regularization in the model.

- Statement 4 calculates the classification report (cr) and returns it to AFS for further processing.

 The *cr* represents several performance metrics such as accuracy, precision, recall, F1-score, and AUC, which ensures the model's efficacy.

- **Accuracy** is the measure of a classifier's ability to make accurate predictions. In balanced classes, accuracy is a valuable metric; nevertheless, it is not a suitable fit in imbalanced classes. Furthermore, the distribution of false positives and negatives is unclear. Accuracy for multiclass classification instances will be calculated using Eq. (4).

$$Accuracy = \frac{\sum_{i=1}^{n} cm[i,j]}{\sum_{i=1}^{n} \sum_{j=1}^{n} cm[i,j]} \tag{4}$$

- **Precision** is helpful when false positives pose a greater risk than false negatives. It is helpful for skewed and unbalanced datasets. As more false positives are predicted by the model, the precision falls. The accuracy of a model's predictions of the target class is guaranteed by its precision. Precision for multiclass classification instances will be calculated using Eq. (5). The classifier's macro-averaged precision score is calculated using the unweighted mean of each class's precision score.

$$Precision_c = \frac{cm[c,c]}{\sum_{i=1}^{n} cm[i,c]} \tag{5}$$

- **Recall** describes the model's ability to predict the real positive cases accurately. When a false positive is more concerning than a false negative, recall is a valuable statistic. Recall decreases with the number of false negatives the model predicts. Recall for multiclass classification instances will be calculated using Eq. (6). The classifier's macro-averaged recall score is calculated using the unweighted mean of each class's recall score.

$$Recall_c = \frac{cm[c,c]}{\sum_{i=1}^{n} cm[c,i]} \tag{6}$$

**Table 2 An illustration of AFS showcasing update in feature weight and improvement in the macro F1-score of the SVC model on VirusShare dataset.** Bold values indicates change in CFWT and corresponding F1-score array iteration wise. The italic values indicate the metrics of best iteration.

| Iteraion | CFWT array [C0,C1,C2, C3, C4,C5,C6,C7] | F1-score array [C0,C1,C2,C3, C4,C5,C6,C7,Macro] | Update CFWT array [C0,C1, C2,C3, C4,C5,C6,C7] | Tighten feature selection criteria of classes | $F_{max}^1$, iteration and Wait$_{cnt}$ |
|---|---|---|---|---|---|
| 1 | [0.0,0.0,0.0,0.0, 0.0,0.0,0.0,0.0] | ['**0.60**','**0.17**','0.91','**0.59**', '0.98', '0.93', '0.96','**0.70**', '**0.73**'] | [**0.01**,**0.01**,0.0,**0.01**, 0.0,0.0,0.0,0.0,**0.01**] | $C_0,C_1,C_3,C_7$ | 0.73,1,5 |
| 2 | [0.01,0.01,0.0,0.0,0.01, 0.0,0.0,0.0,0.0,0.01] | ['**0.59**', '**0.17**','0.91','**0.59**', '0.98', '0.94', '0.96', '**0.71**', '**0.73**'] | [**0.02**,**0.02**,0.0,**0.02**, 0.0,0.0,0.0,0.0,**0.02**] | $C_0,C_1,C_3,C_7$ | 0.73,2,5 |
| 3 | [0.02,0.02,0.0,0.0,0.02, 0.0,0.0,0.0,0.0,0.02] | ['**0.62**','**0.17**','0.92','**0.59**', '0.98', '0.94', '0.96', '**0.70**', '**0.73**'] | [**0.03**,**0.03**,0.0,**0.03**, 0.0,0.0,0.0,0.0,**0.03**] | $C_0,C_1,C_3,C_7$ | 0.73,3,5 |
| 4 | [0.03,0.03,0.0,0.0,0.03, 0.0,0.0,0.0,0.0,0.03] | ['**0.64**', '**0.26**','0.91','**0.59**', '0.98', '0.94', '0.96', '**0.71**', '**0.75**'] | [**0.04**,**0.04**,0.0,**0.04**, 0.0,0.0,0.0,0.0,**0.04**] | $C_0,C_1,C_3,C_7$ | 0.75,4,5 |
| 5 | [0.04,0.04,0.0,0.0,0.04, 0.0,0.0,0.0,0.0,0.04] | ['0.84', '**0.31**', '0.92', '0.84', '0.98', '0.95', '0.96', '**0.72**', '**0.81**'] | [0.04,**0.05**,0.0,0.04, 0.0,0.0,0.0,0.0,**0.05**] | $C_1,C_7$ | 0.81,5,5 |
| 6 | [0.04,0.05,0.0,0.0,0.04, 0.0,0.0,0.0,0.0,0.05] | ['**0.85**', '**0.81**','0.92','**0.67**', '0.98', '0.96', '0.96', '**0.73**', '**0.86**'] | [**0.05**,**0.06**,0.0,**0.05**, 0.0,0.0,0.0,0.0,**0.06**] | $C_0,C_1,C_3,C_7$ | 0.86,6,5 |
| 7 | [0.05,0.06,0.0,0.0,0.05, 0.0,0.0,0.0,0.0,0.06] | ['**0.86**', '**0.75**','0.91','**0.84**', '0.98', '0.96', '0.96', '**0.72**', '**0.87**'] | [**0.06**,**0.07**,0.0,**0.06**, 0.0,0.0,0.0,0.0,**0.07**] | $C_0,C_1,C_3,C_7$ | 0.87,7,5 |
| *8 | [*0.06,0.07,0.0,0.0,0.06, 0.0,0.0,0.0,0.0,0.07*] | ['*0.91*', '*0.89*','*0.92*', '*0.93*', '*0.98*', '*0.96*', '*0.96*', '*0.72*', '*0.91*'] | [*0.07*,*0.08*,0.0,0.0,0.06, 0.0,0.0,0.0,0.0,*0.08*] | $C_0,C_1,C_7$ | *0.91,8,5* |
| 9 | [0.07,0.08,0.0,0.0,0.06, 0.0,0.0,0.0,0.0,0.08] | ['0.90','**0.36**', '0.91', '0.89', '0.98', '0.96', '0.96', '**0.70**', '**0.83**'] | [0.07,**0.09**,0.0,0.0,0.06, 0.0,0.0,0.0,0.0,**0.09**] | $C_1,C_7$ | 0.91,8,4 |
| 10 | [0.07,0.09,0.0,0.0,0.06, 0.0,0.0,0.0,0.0,0.09] | ['0.91', '**0.38**', '0.92', '0.95', '0.98', '0.96', '0.96', '**0.70**', '**0.84**'] | [0.07,**0.10**,0.0,0.0,0.06, 0.0,0.0,0.0,0.0,**0.10**] | $C_1,C_7$ | 0.91,8,3 |
| 11 | [0.07,0.10,0.0,0.0,0.06, 0.0,0.0,0.0,0.0,0.10] | ['0.92', '**0.37**', '0.92', '0.93', '0.98', '0.96', '0.96', '**0.75**', '**0.85**'] | [0.07,**0.11**,0.0,0.0,0.06, 0.0,0.0,0.0,0.0,**0.11**] | $C_1,C_7$ | 0.91,8,2 |
| 12 | [0.07,0.11,0.0,0.0,0.06, 0.0,0.0,0.0,0.0,0.11] | ['0.90', '**0.36**', '0.91', '0.88', '0.98', '0.96', '0.96', '**0.79**', '**0.84**'] | [0.07,**0.12**,0.0,0.0,0.06, 0.0,0.0,0.0,0.0,**0.12**] | $C_1,C_7$ | 0.91,8,1 |
| 13 | [0.07,0.12,0.0,0.0,0.06, 0.0,0.0,0.0,0.0,0.12] | ['0.92', '**0.37**', '0.92', '0.91', '0.98', '0.96', '0.96', '**0.78**', '**0.85**'] | [0.07,**0.13**,0.0,0.0,0.06, 0.0,0.0,0.0,0.0,**0.13**] | $C_1,C_7$ | 0.91,8,0 |
| 14 | [0.07,0.13,0.0,0.0,0.06, 0.0,0.0,0.0,0.0,0.13] | ['0.91', '**0.38**', '0.91', '0.92', '0.98', '0.96', '0.96', '**0.78**', '**0.85**'] | [0.07,**0.14**,0.0,0.0,0.06, 0.0,0.0,0.0,0.0,**0.14**] | $C_1,C_7$ | 0.91,8,−1 STOP |

**Note:**

\* Best iteration.

- The weighted average of recall and precision is known as the F1-score. Recall and precision must both be strong for the classifier to have a high F1-score. This measure only gives preference to classifiers with comparable recall and precision. F1-score for multiclass classification instances will be calculated using Eq. (7). The classifier's macro-averaged F1-score is calculated using the unweighted mean of each class's F1-score.

$$F_{1c} = \frac{2}{\frac{1}{Recall_c} + \frac{1}{Precision_c}} \tag{7}$$

Table 2 provides an example of AFS demonstrating the modification of the selection criteria and enhancement of the macro F1-score of the SVC model on the VirusShare

**Table 3 Description of dataset.**

| (A) Imbalanced class | | | | (B) Imbalanced feature sequence | | | | | | |
|---|---|---|---|---|---|---|---|---|---|---|
| Malware class | VirusSample | VirusShare | MAL-API-2019 | API sequence length | VirusSample | | VirusShare | | MAL-API-2019 | |
| | | | | | Original | Dup_Rem | Original | Dup_Rem | Original | Dup_Rem |
| Virus | 2,367 | 2,490 | 1,001 | >0&<20 | 6,821 | 6,821 | 7,458 | 7,458 | 55 | 1,227 |
| Trojan | 6,153 | 8,919 | 1,001 | >=20&<40 | 331 | 335 | 473 | 478 | 281 | 2,316 |
| Worms | 441 | 524 | 1,001 | >=40&<60 | 470 | 490 | 600 | 619 | 294 | 2,241 |
| Backdoor | 447 | 510 | 1,001 | >=60&<80 | 116 | 127 | 434 | 455 | 191 | 845 |
| Adware | 222 | 908 | 379 | >=80&<100 | 275 | 323 | 717 | 819 | 237 | 372 |
| Downloader | NA | 218 | 1,001 | >=100&<150 | 1,118 | 1,037 | 1,616 | 1,594 | 316 | 106 |
| Agent | 102 | 165 | NA | >=150&<200 | 66 | 70 | 746 | 627 | 270 | 0 |
| Ransomware | NA | 115 | NA | >=200&<250 | 172 | 167 | 902 | 897 | 267 | 0 |
| Spyware | NA | NA | 832 | >=250&<300 | 290 | 289 | 408 | 409 | 263 | 0 |
| Dropper | NA | NA | 891 | >=300 | 73 | 73 | 495 | 493 | 4,933 | 0 |
| Total sample size | 9,732 | 13,849 | 7,107 | Total no. of API sequences | 9,732 | 9,732 | 13,849 | 13,849 | 7,107 | 7,107 |

dataset. The greedy approach in updating the feature selection criteria over iterations is shown in this table. Since the goal is to improve the model's performance for an imbalanced multiclass problem, the greedy property ensures improvement in the macro F1-score of the model by improving the F1-score of the classes with poor performance. Here, the greedy approach updates *CFWT[]* while considering the F1[] score array from the model's cr. The primary step of the technique is only to update a class's feature selection threshold criteria in *CFWT[]* when the class's F1-score is less than or equal to the model's macro F1-score.

In every iteration, AFS compares $F_1^t$ with the macro F1-score of the model. The initial value of $F_1^t$ for the first iteration in Table 2 is 0.75, which is greater than the model's macro F1-score of 0.73. The fourth iteration resets $F_1^t$ to 0.80, which is attained in the fifth. Moreover, the model's $F_1^{max}$ score is determined to be 0.91 in iteration 8, and $F_1^t$ is reset to 0.95. AFS ends after the fourteenth iteration since the model's macro F1-score has not increased during the previous six iterations. Whenever the model detects a macro F1-score higher than or equal to the most recent known $F_1^{max}$, the parameter $Wait_{count}$ returns to its initial value, and $F_1^{max}$ gets updated. When the macro F1-score does not exceed the most recent $F_1^{max}$, the $Wait_{count}$ decreases to prevent AFS from nonterminating. After the fourteenth iteration, the search for a higher macro F1-score ended since the parameter $Wait_{count}$, which has an initial value of 5, expired. Finally, AFS returned the $F_1^{max}$ of 0.91, CFWT array [0.07,0.08,0.0,0.0,0.06,0.0,0.0,0.0,0.0,0.08] of eight classes and iteration number 8 representing the classification report of the iteration with the maximum macro F1-score.

## EXPERIMENTAL EVALUATION

This section highlights the datasets used to evaluate the proposed framework in terms of its efficacy with several performance metrics. The experimental work for the described

**Table 4** Ten-fold cross validation performance measures of AMMC for VirusShare.

| | (A) Without AFS | | | | (B) With AFS ($F_t^l = 0.95$ and $W_{cnt} = 5$) | | | |
| --- | --- | --- | --- | --- | --- | --- | --- | --- |
| | SVC | RF | KNN | ECLF | SVC | RF | KNN | ECLF |
| Precision | 0.88 ± 0.01 | 0.90 ± 0.01 | 0.84 ± 0.02 | 0.89 ± 0.03 | 0.94 ± 0.01 | 0.97 ± 0.00 | 0.93 ± 0.01 | 0.95 ± 0.01 |
| Recall | 0.69 ± 0.00 | 0.68 ± 0.01 | 0.71 ± 0.02 | 0.64 ± 0.03 | 0.87 ± 0.01 | 0.87 ± 0.01 | 0.88 ± 0.00 | 0.81 ± 0.03 |
| F1 | 0.73 ± 0.00 | 0.73 ± 0.00 | 0.75 ± 0.01 | 0.70 ± 0.02 | 0.90 ± 0.01 | 0.92 ± 0.01 | 0.91 ± 0.00 | 0.87 ± 0.02 |
| Accuracy | 0.90 ± 0.00 | 0.90 ± 0.01 | 0.90 ± 0.00 | 0.90 ± 0.01 | 0.95 ± 0.00 | 0.95 ± 0.00 | 0.94 ± 0.00 | 0.94 ± 0.01 |

**Table 5** Ten-fold cross validation performance measures of AMMC for VirusSample.

| | (A) Without AFS | | | | (B) With AFS ($F_t^l = 0.95$ and $W_{cnt} = 5$) | | | |
| --- | --- | --- | --- | --- | --- | --- | --- | --- |
| | SVC | RF | KNN | ECLF | SVC | RF | KNN | ECLF |
| Precision | 0.94 ± 0.03 | 0.93 ± 0.07 | 0.88 ± 0.04 | 0.93 ± 0.04 | 0.95 ± 0.02 | 0.97 ± 0.01 | 0.95 ± 0.01 | 0.96 ± 0.01 |
| Recall | 0.71 ± 0.02 | 0.72 ± 0.03 | 0.75 ± 0.05 | 0.72 ± 0.02 | 0.87 ± 0.03 | 0.87 ± 0.03 | 0.86 ± 0.03 | 0.87 ± 0.03 |
| F1 | 0.76 ± 0.02 | 0.78 ± 0.03 | 0.78 ± 0.02 | 0.76 ± 0.02 | 0.90 ± 0.02 | 0.92 ± 0.02 | 0.90 ± 0.02 | 0.91 ± 0.02 |
| Accuracy | 0.94 ± 0.01 | 0.95 ± 0.00 | 0.94 ± 0.01 | 0.94 ± 0.01 | 0.96 ± 0.01 | 0.96 ± 0.01 | 0.96 ± 0.01 | 0.96 ± 0.00 |

**Table 6** Ten-fold cross validation performance measures of AMMC for MAL-API-2019.

| | (A) Without AFS | | | | (B) With AFS ($F_t^l = 0.85$ and $W_{cnt} = 15$) | | | |
| --- | --- | --- | --- | --- | --- | --- | --- | --- |
| | SVC | RF | KNN | ECLF | SVC | RF | KNN | ECLF |
| Precision | 0.65 ± 0.01 | 0.66 ± 0.01 | 0.62 ± 0.01 | 0.66 ± 0.01 | 0.81 ± 0.02 | 0.79 ± 0.01 | 0.73 ± 0.02 | 0.80 ± 0.01 |
| Recall | 0.65 ± 0.01 | 0.64 ± 0.01 | 0.62 ± 0.01 | 0.65 ± 0.01 | 0.80 ± 0.02 | 0.77 ± 0.01 | 0.74 ± 0.01 | 0.79 ± 0.01 |
| F1 | 0.65 ± 0.01 | 0.65 ± 0.01 | 0.61 ± 0.01 | 0.65 ± 0.01 | 0.81 ± 0.02 | 0.78 ± 0.01 | 0.73 ± 0.01 | 0.79 ± 0.01 |
| Accuracy | 0.64 ± 0.01 | 0.64 ± 0.01 | 0.61 ± 0.01 | 0.64 ± 0.01 | 0.81 ± 0.02 | 0.77 ± 0.01 | 0.73 ± 0.01 | 0.79 ± 0.01 |

AMMC is carried out using an Intel(R) Core(TM) i5-1035G1 CPU @ 1.00GHz X 8 PC with 16 GB of RAM, Ubuntu-22.04.5 LTS, and Python 3.9.

## Dataset description

The experiment was conducted using three benchmark multiclass malware datasets, as shown in Table 3, to demonstrate the efficacy of AMMC. The multiclass dataset MAL-API-2019 contains 7,107 dynamically collected API call sequences from eight types of malware on the Windows operating system. This dataset, which includes sequential API calls, enables researchers to comprehend how metamorphic malware modifies its behavior (*i.e.*, API calls) by appending opcodes that have no significance. VirusSample and VirusShare datasets contain 9,795 and 14,616 statically collected API call sequences extracted from six and eight types of malware portable executable (PE) file headers, respectively. During the experiment, all malware classes in VirusSample and VirusShare

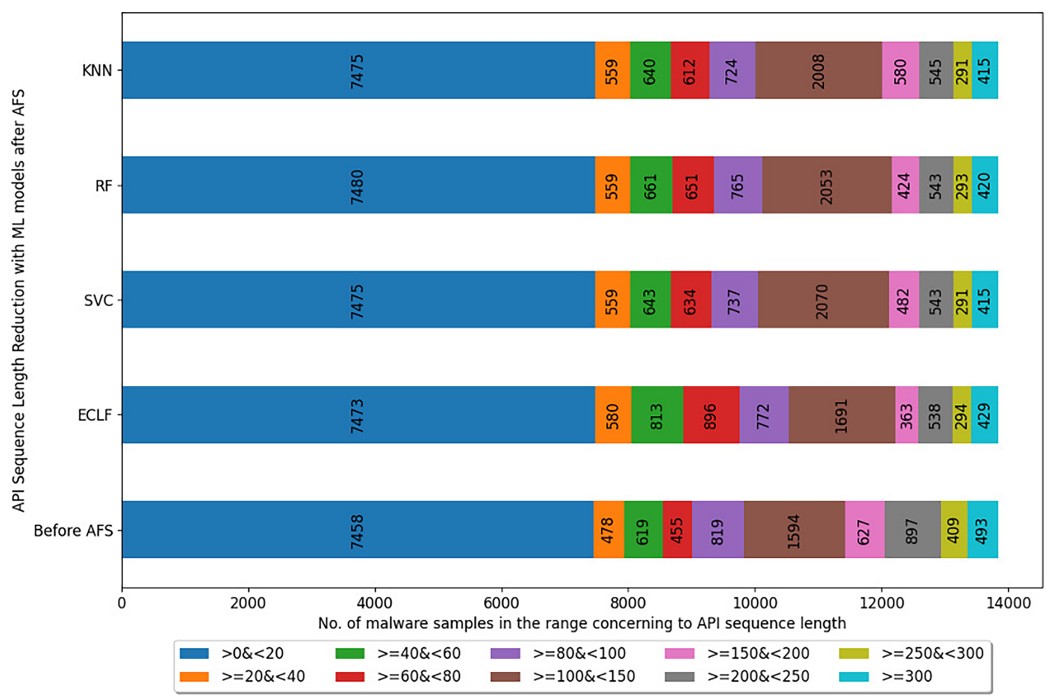

**Figure 5  Reduced length API sequences at iteration with maximal macro F1 for VirusShare with AFS.**

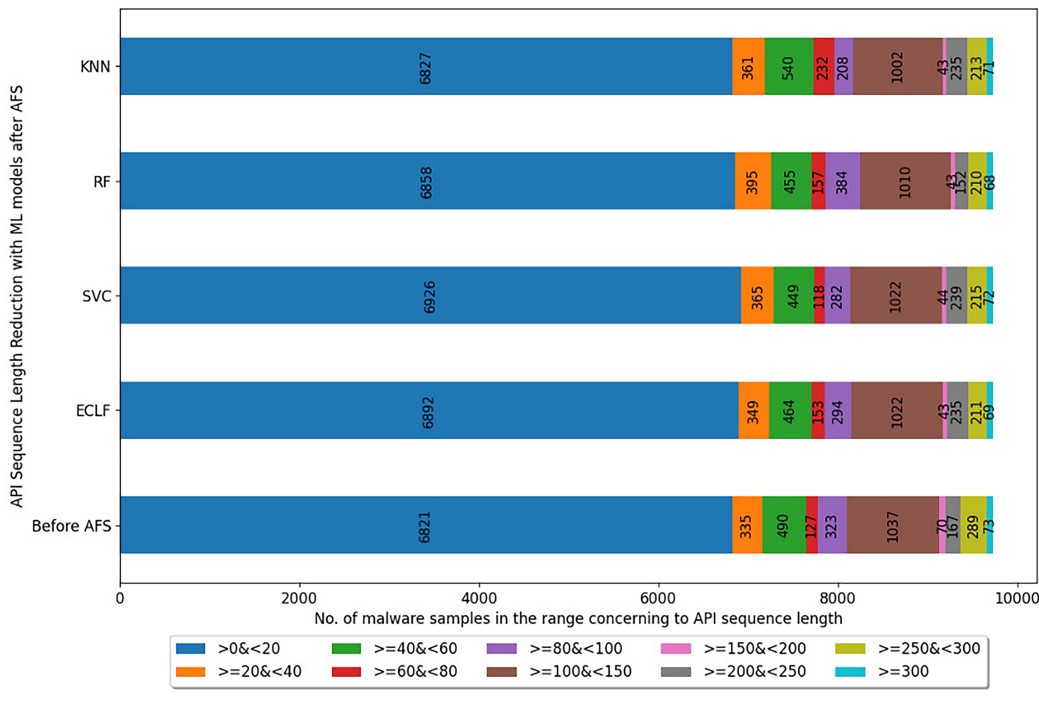

**Figure 6  Reduced length API sequences at iteration with maximal macro F1 for VirusSample with AFS.**

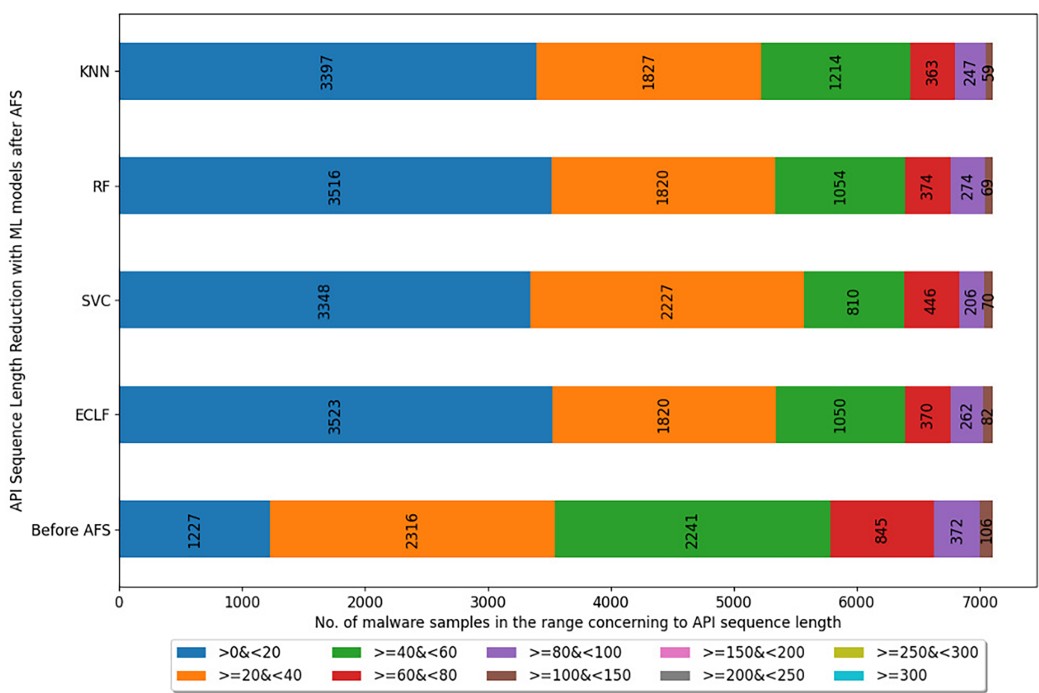

**Figure 7 Reduced length API sequences at iteration with maximal macro F1 for MAL-API-2019 with AFS.**

with more than 100 samples were considered; however, MAL-API-2019 was considered thoroughly. The sample sizes of different classes outlined in Table 3A illustrate the class-wise data imbalance issue. Furthermore, all three datasets contain API sequences of varying lengths. Considering an API request as a feature of a sample, such diversity in API sequence length represents feature-wise imbalance. Table 3B depicts the variation in API sequence length concerning the frequency of such API sequences in all datasets before and after the duplicate API's elimination in each sequence. During the experiment, AMMC handled the issue of feature sequence imbalance by preserving the class-wise sample imbalance issue.

## Result analysis

Three fundamental machine learning models, RF, KNN, and SVC, are employed for training and testing in order to guarantee the goal of validating AMMC with models that use fewer computational resources. KNN is used as one of the supervised non-linear machine learning models on vector distance logic, RF is used as an implicit ensemble model, and SVC is used as a candidate linear and non-linear model that can operate on diverse datasets. Additionally, ECLF is employed as an ensemble classifier, with the above three being estimators based on soft voting.

The hyperparameters (WindowSize=10, VectorSize=200) are set through a random search to estimate API weight vectors using the Skip-gram model. Furthermore, the API weight vectors are normalized using the Standard Scalar after considering multiple

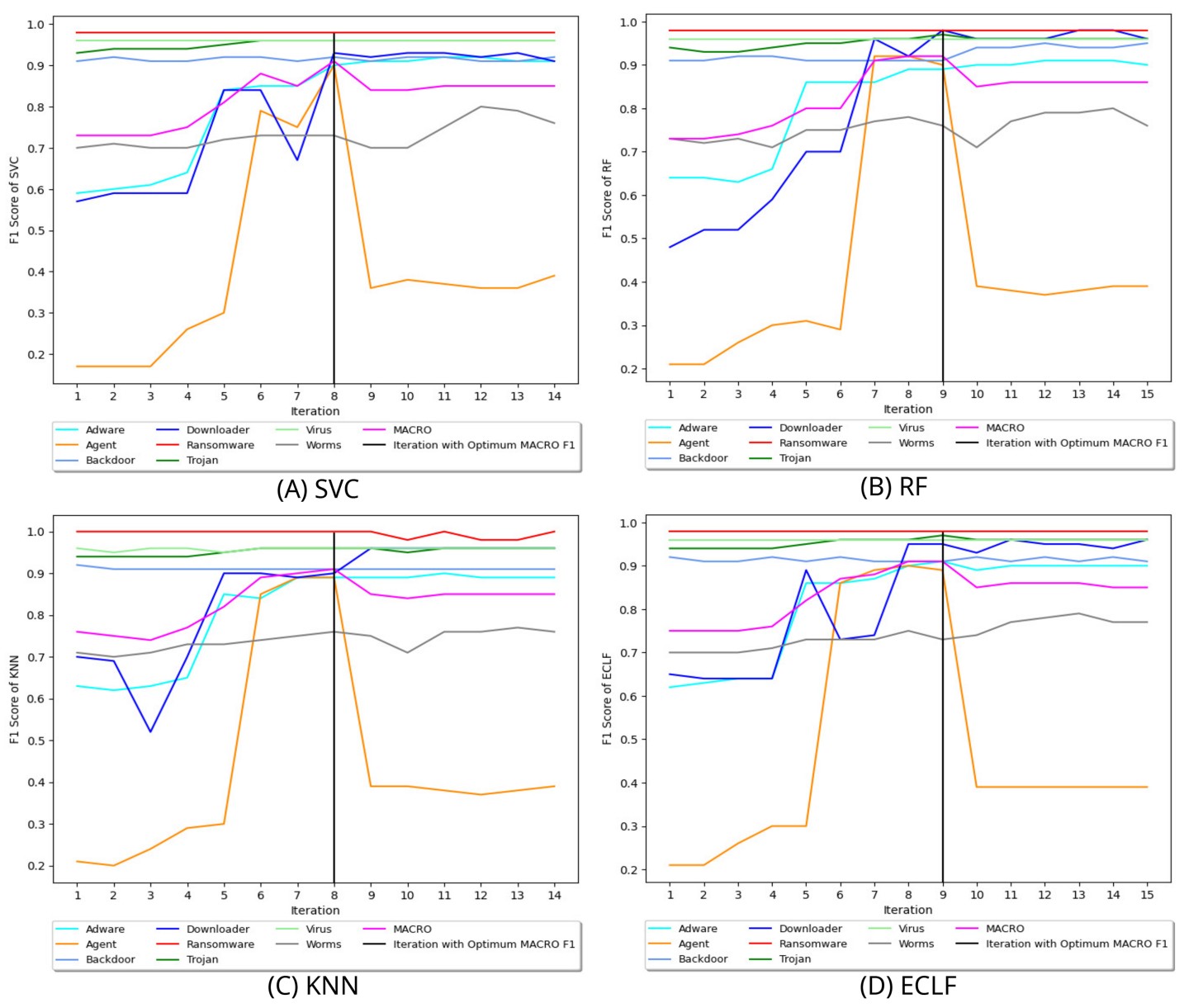

**Figure 8** (A–D) Change in $F_1^{max}$ in search of maximal macro F1 with $F_1^t = 0.95$ and $W_{cnt} = 5$ for VirusShare.

normalization techniques such as MinMax, Robust, and Standard Scalar. Through the grid search technique, emphasizing regularization, the hyperparameters for SVC, RF, and KNN are determined to be (gamma='auto', kernel='rbf', C=10, max_iter=10000), (n_estimators=200, criteria = 'entropy', bootstrap=True, max_depth=1000), and (n_neighbors=8, weights='distance', algorithm='auto', leaf_size=30, p=2, metric='minkowski'), respectively.

An ablation study is carried out on the three benchmark datasets by dropping the stage *Select Feature by Class* and the *Greedy Strategy for updating feature weights* in AMMC to

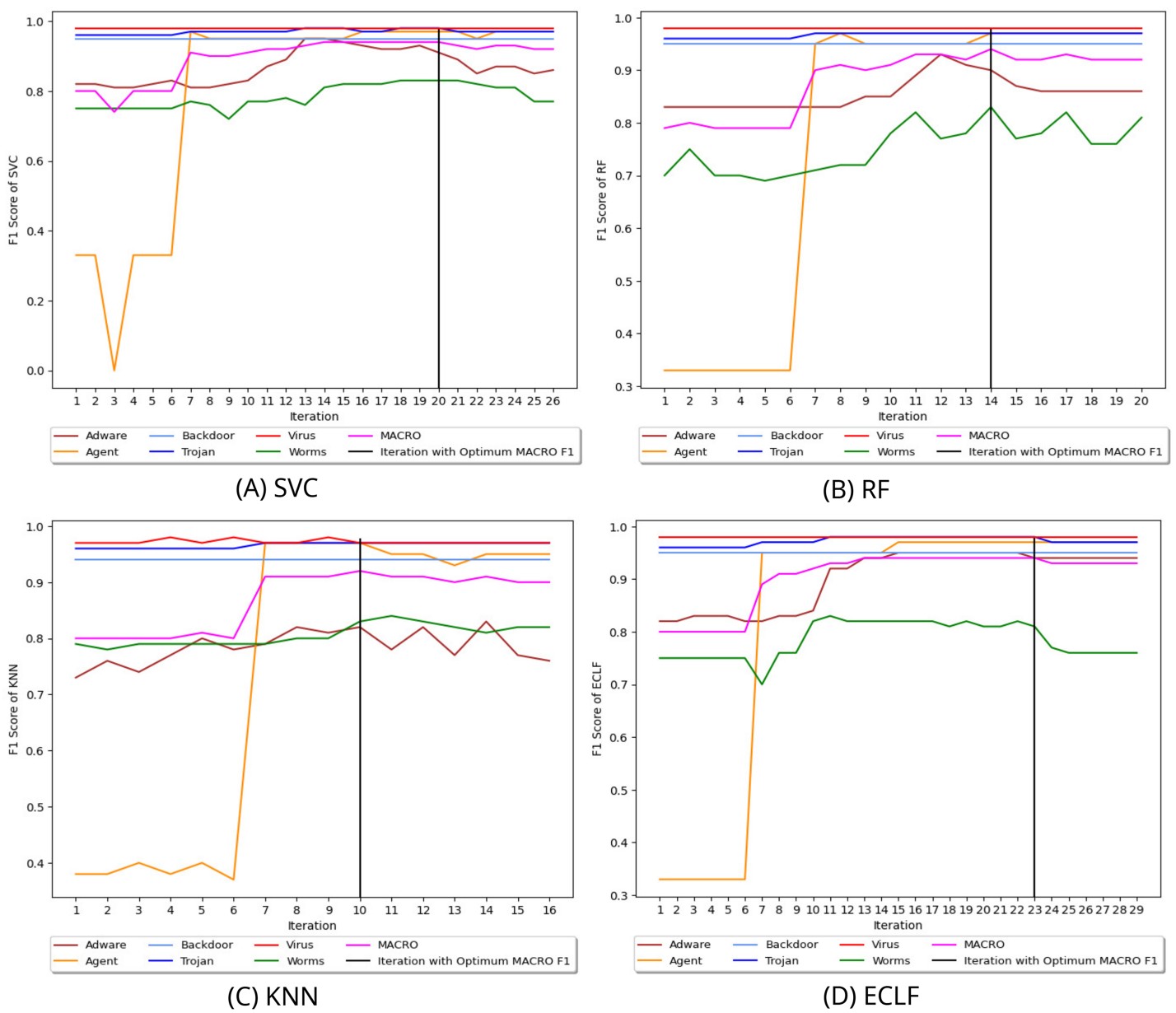

**Figure 9** (A–D) Change in $F_1^{max}$ in search of maximal macro F1 with $F_1^t = 0.95$ and $W_{cnt} = 5$ for VirusSample.

ensure the impact of AFS. This experiment on AMMC without AFS uses the API sequences generated by the *Duplicate API Removal* stage to train and validate classifiers. The mean and standard deviation of 10-fold cross-validation results for macro precision, macro recall, macro F1-score, and accuracy are displayed in Tables 4A, 5A and 6A respectively.

To validate AMMC's efficacy, we reran the experiment on the three benchmark datasets with AFS. The AFS-Pruned dataset is obtained *via SelectFeatureByClass*. To stress

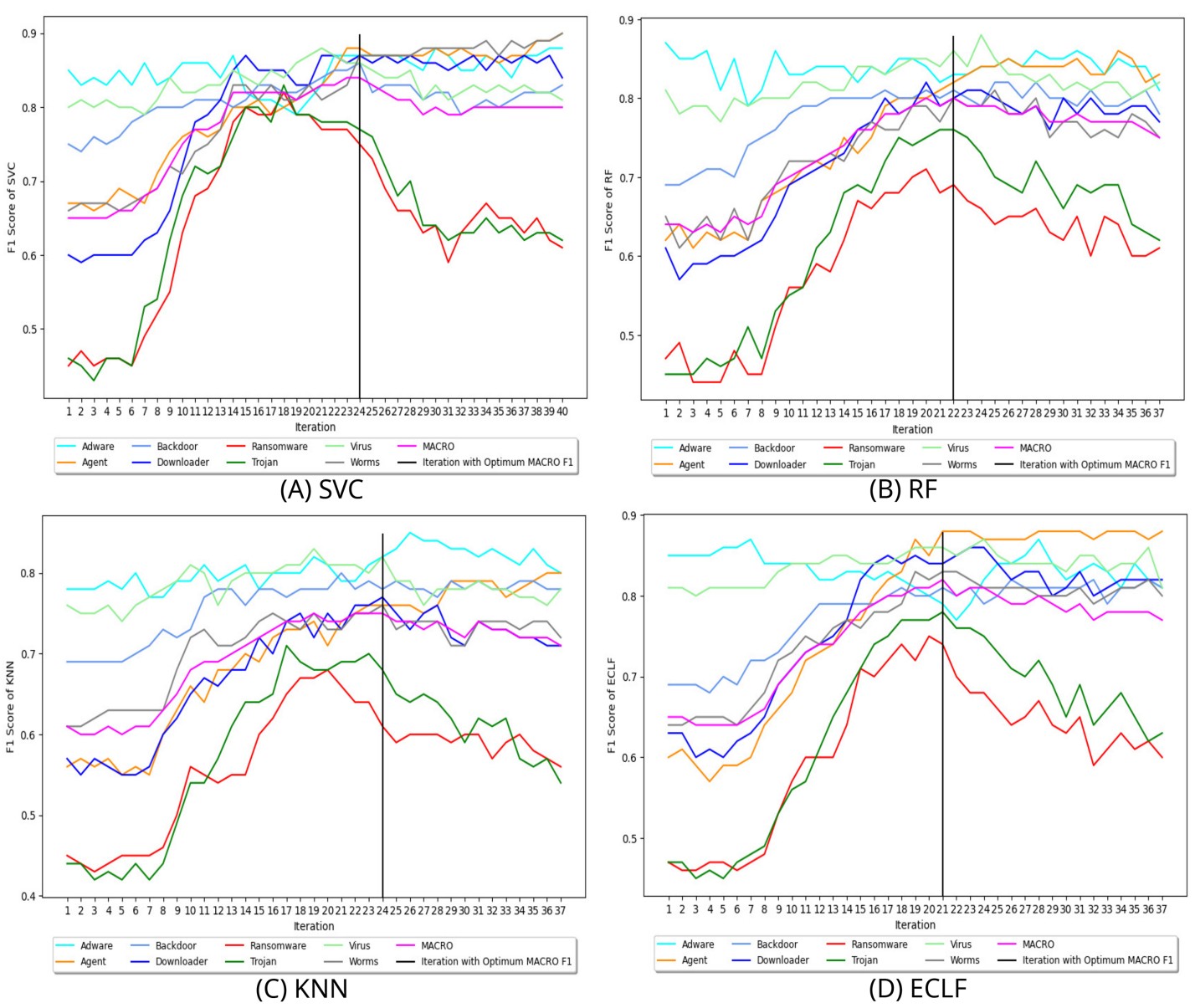

**Figure 10 (A–D)** Change in $F_1^{max}$ in search of maximal macro F1 with $F_1^t = 0.85$ and $W_{cnt}$ = 15 for MAL-API-2019.

consistency in performance indicators, each model undergoes a 10-fold cross-validation. AFS goes through multiple iterations in each fold to get the maximal macro F1-score ($F_1^{max}$) by training and evaluating the basic machine learning and ensemble models outlined before. Compared to the macro F1-score without AFS, a higher macro F1 target ($F_1^t$) and $Wait_{cnt}$ are gradually adjusted to seek a superior macro F1-score with AFS. Tables 4B, 5B, and 6B exhibit the 10-fold cross-validation findings, including the mean and standard deviation of maximal macro F1-scores for each fold. Furthermore, the target macro F1 ($F_1^t$) and the related $Wait_{cnt}$ are superscribed in the Tables 4B, 5B, and 6B,

**Table 7 Maximal performance metrics of VirusShare.** Bold values indicate the best performance.

| Models | Fold | Iteration | Feature weight threshold array CFWT [1..n] for maximal macro F1 | | | | | | | | Macro F1 | Macro ROC-AUC | MCC | MTTD (ms) Test size 2,770 |
|---|---|---|---|---|---|---|---|---|---|---|---|---|---|---|
| | | | Adware | Agent | Backdoor | Downloader | Ransomware | Trojan | Virus | Worms | | | | |
| SVC | 5 | 8 | 0.06 | 0.07 | 0.00 | 0.06 | 0.00 | 0.00 | 0.00 | 0.07 | 0.91 | 0.98 | 0.90 | 0.330 |
| RF | 1 | 9 | 0.06 | 0.07 | 0.02 | 0.07 | 0.00 | 0.00 | 0.00 | 0.08 | **0.92** | **0.99** | **0.91** | 0.030 |
| KNN | 2 | 8 | 0.06 | 0.07 | 0.00 | 0.05 | 0.00 | 0.00 | 0.00 | 0.07 | 0.91 | 0.97 | 0.90 | 0.247 |
| ECLF | 8 | 9 | 0.07 | 0.07 | 0.01 | 0.06 | 0.00 | 0.00 | 0.00 | 0.08 | 0.91 | 0.99 | 0.91 | 0.760 |

**Table 8 Maximal performance metrics of VirusSample.** Bold values indicate the best performance.

| Models | Fold | Iteration | Feature weight threshold array CFWT [1..n] for maximal macro F1 | | | | | | Macro F1 | Macro ROC-AUC | MCC | MTTD (ms) Test size 1,947 |
|---|---|---|---|---|---|---|---|---|---|---|---|---|
| | | | Adware | Agent | Backdoor | Trojan | Virus | Worms | | | | |
| SVC | 2 | 20 | 0.11 | 0.06 | 0.00 | 0.00 | 0.00 | 0.18 | 0.94 | 0.98 | 0.94 | 0.146 |
| RF | 2 | 14 | 0.07 | 0.06 | 0.00 | 0.00 | 0.00 | 0.13 | **0.94** | **0.99** | **0.94** | 0.024 |
| KNN | 3 | 10 | 0.09 | 0.06 | 0.00 | 0.00 | 0.00 | 0.09 | 0.92 | 0.98 | 0.92 | 0.186 |
| ECLF | 2 | 23 | 0.08 | 0.06 | 0.00 | 0.00 | 0.00 | 0.22 | 0.94 | 0.99 | 0.94 | 0.701 |

**Table 9 Maximal performance metrics of MAL-API-2019.** Bold values indicate the best performance.

| Models | Fold | Iteration | Feature weight threshold array CFWT[1..n] for maximal macro F1 | | | | | | | | Macro F1 | Macro ROC-AUC | MCC | MTTD (ms) Test size 1,422 |
|---|---|---|---|---|---|---|---|---|---|---|---|---|---|---|
| | | | Adware | Backdoor | Downloader | Dropper | Spyware | Trojan | Virus | Worms | | | | |
| SVC | 9 | 24 | 0.07 | 0.12 | 0.03 | 0.10 | 0.23 | 0.22 | 0.00 | 0.14 | **0.84** | **0.98** | **0.82** | 1.143 |
| RF | 7 | 22 | 0.03 | 0.14 | 0.05 | 0.12 | 0.21 | 0.21 | 0.00 | 0.08 | 0.80 | 0.97 | 0.77 | 0.040 |
| KNN | 9 | 24 | 0.05 | 0.14 | 0.07 | 0.12 | 0.23 | 0.23 | 0.00 | 0.08 | 0.75 | 0.94 | 0.74 | 0.361 |
| ECLF | 9 | 21 | 0.02 | 0.14 | 0.04 | 0.12 | 0.20 | 0.20 | 0.00 | 0.08 | 0.82 | 0.98 | 0.79 | 1.872 |

together with the macro precision, recall, and accuracy. Considering all the classifier's performance, the impact of AMMC with AFS in contrast to AMMC without AFS is well justified by showing significant improvement with the several performance metrics outlined in Tables 4–6.

Figures 5–7 depicts the reduction in API sequence length achieved by selecting APIs that meet the weight criterion *via SelectFeatureByClass* for all models on all datasets during the fold, with the highest macro F1-score with the previously indicated *macro $F_1^t$*. In the dataset found after *Duplicate API Removal* stage, there were 493 API sequences of length greater than 300 for *VirusShare*, as shown in Fig. 5. However, using *SelectFeatureByClass* in *AFS* lowered the number of API sequences with lengths >=300 from 493 to 429, 415, 420,

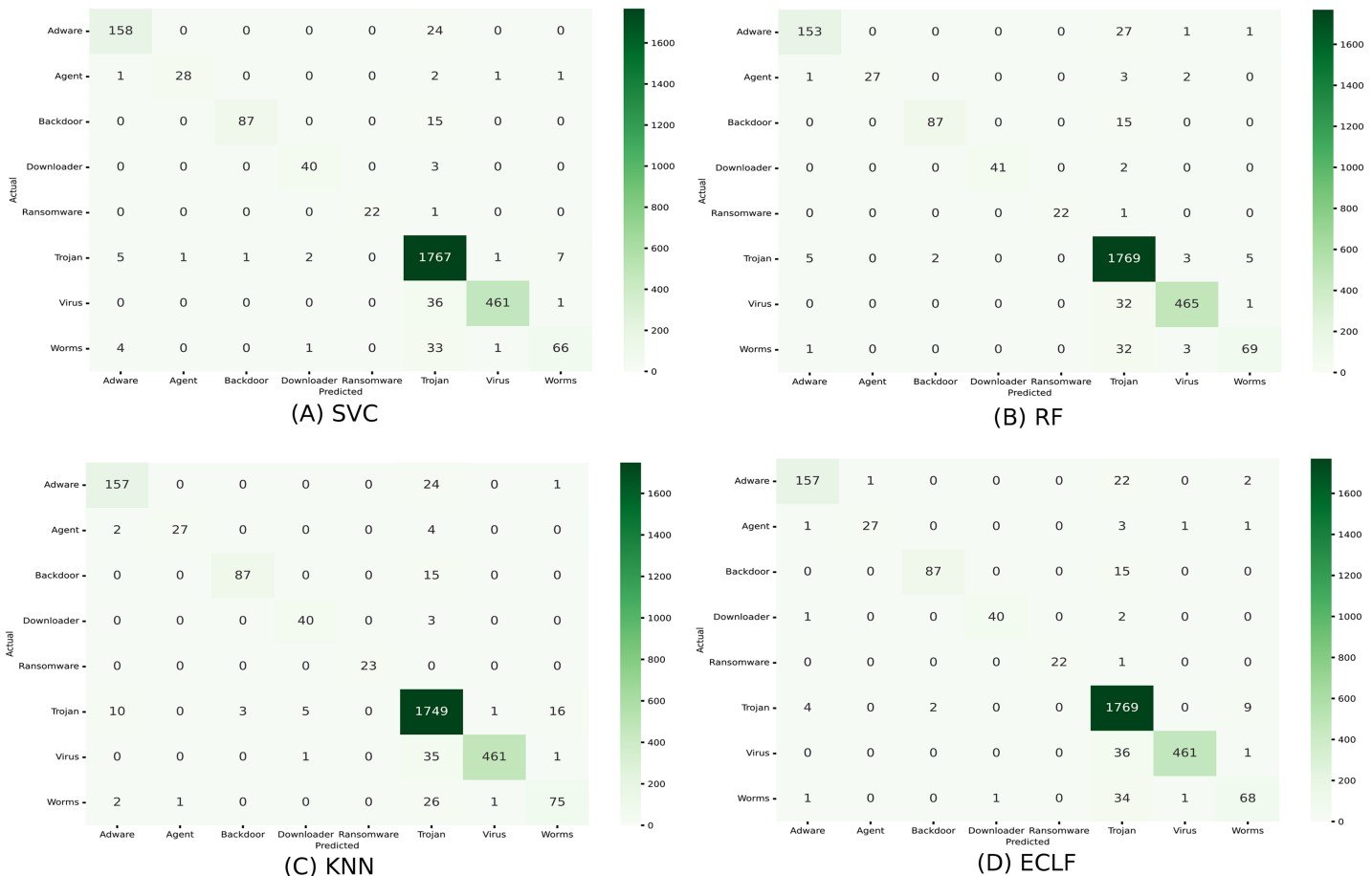

**Figure 11 (A–D) Confusion matrix of models for the iteration with maximal macro F1 of VirusShare.**

and 415, respectively, for the evaluated models. As a result, the number of API sequences increases in the range $>=100\&<150, >=60\&<80, >=40\&<60, >=20\&<40,$ and $>0\&<20,$ respectively. It is observed that in the dataset formed after *Duplicate API Removal* stage, longer API sequences are becoming shorter and shifting toward the lower-length API sequences. Figures 6 and 7 show similar observations for the *VirusSample* and *MAL-API-2019* datasets, respectively. With the reduced API sequences referring to the mean accuracy, precision, recall, and F1-scores for all three datasets highlighted in Tables 4–6 the effect of AFS is well established.

Figure 8 depicts the process of iteratively searching superior macro F1 at the fold with the highest macro F1 for all models using VirusShare. We stopped our experiment at $F_1^t = 0.95$ and $W_{cnt} = 5$. In the eighth iteration, both SVC and KNN achieved the $F_1^{max}$ score of 0.91 and ended the quest for a higher macro F1-score in iteration 14. However, RF and ECLF achieved the $F_1^{max}$ score of 0.92 and 0.91 at iteration 9 and ended their pursuit for a higher macro F1-score in iteration 15. Similarly, Figs. 9 and 10 depict iteratively searching for superior macro F1 at the fold with the highest macro F1 for all models using VirusSample and MAL-API-2019.

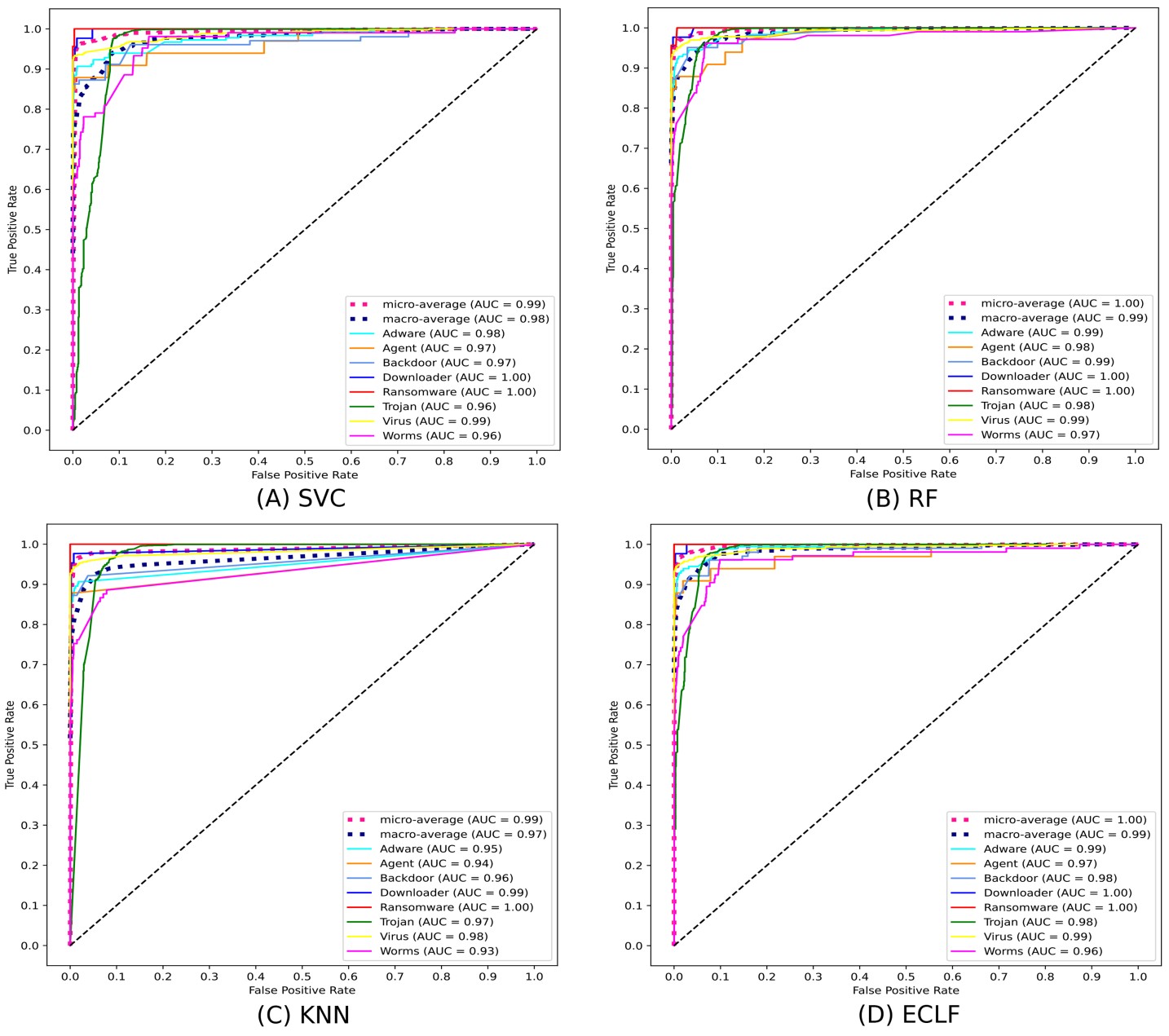

**Figure 12  (A–D) ROC-AUC of models for the iteration with maximal macro F1 of VirusShare.**

Table 7 highlights the maximum macro F1-score and macro ROC-AUC achieved on *VirusShare* for all the models with their respective fold and iteration in that fold. One statistical method used to assess a model's ability to predict multiclass classifications is the Matthews correlation coefficient (MCC). It is employed in machine learning to assess how well predictions are made. On unbalanced datasets, it is more trustworthy than accuracy and F1-score, which might be deceptive. Because it considers the balance of the four

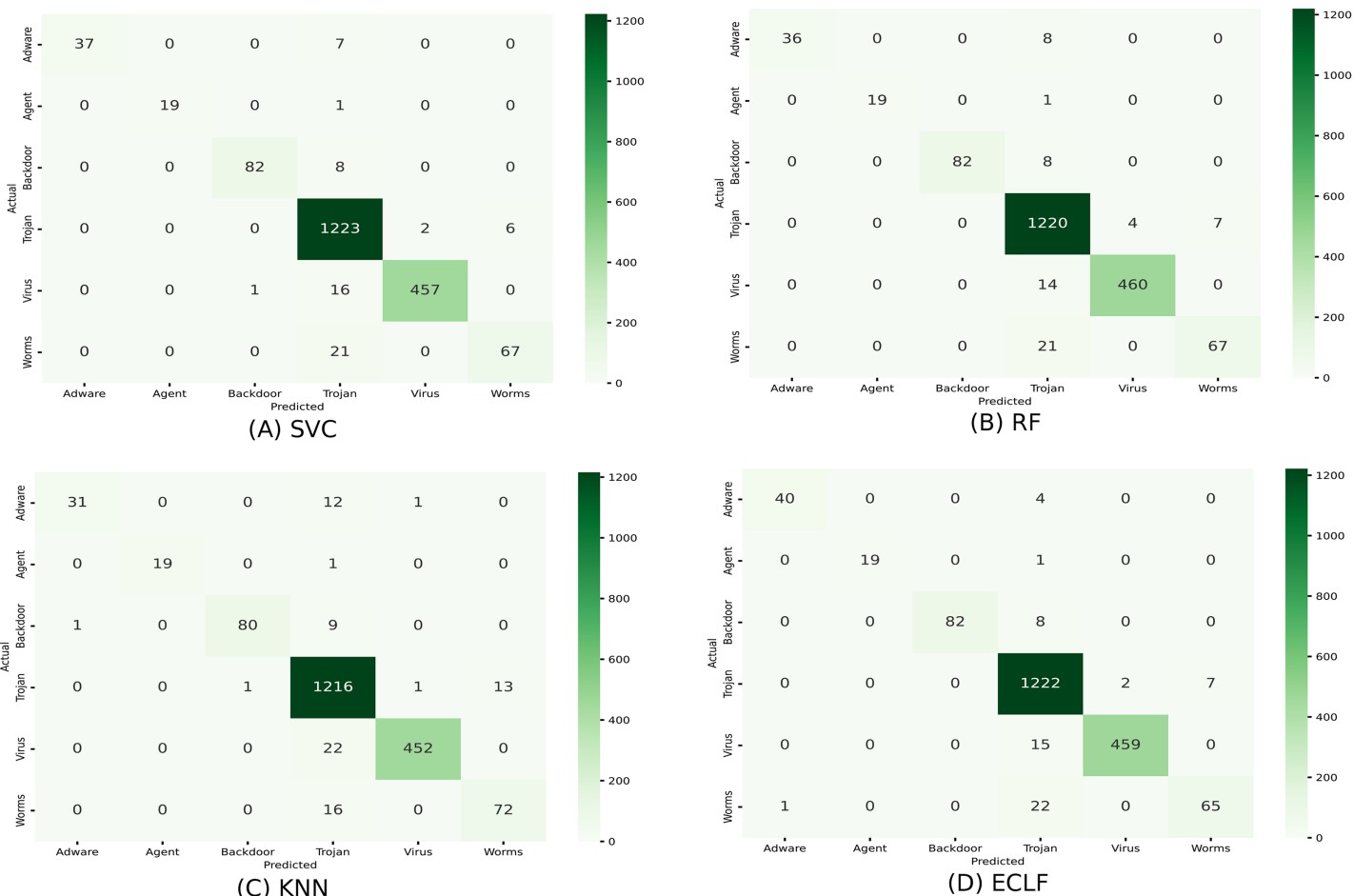

**Figure 13 (A–D) Confusion matrix of models for the iteration with maximal macro F1 of VirusSample.**

confusion matrix entries (TP, TN, FP, and FN), it is more informative than accuracy and F1-score. Since class sizes vary in all the datasets, the MCC scores for each model are also shown in this table to guarantee a fair evaluation. Furthermore, the mean time to detect (MTTD) in milliseconds shows the detection time efficiency in this multiclass problem. During the search for the $F_1^{max}$ score *via* AFS, the F1-score for classes *Ransomware, Trojan, and Virus* always stands higher than the macro F1-score of the model. Hence, *SelectFeatureByClass* method does not make any changes to their API sequences. However, for all other classes, the API selection in a sequence is done by comparing the respective class weight threshold in CFWT *via SelectFeatureByClass*. The contents of CFWT for the iteration with the maximum macro F1-score are also shown in the table. Tables 8 and 9 represent the CFWT, MCC, and MTTD details of the fold and iteration with maximum macro F1 concerning datasets *VirusSample* and *MAL-API-2019*.

Figures 11 and 12 show the confusion matrix and ROC curve with AUC scores for all the models on VirusShare of the fold with maximum macro F1-score. Similarly, Figs. 13

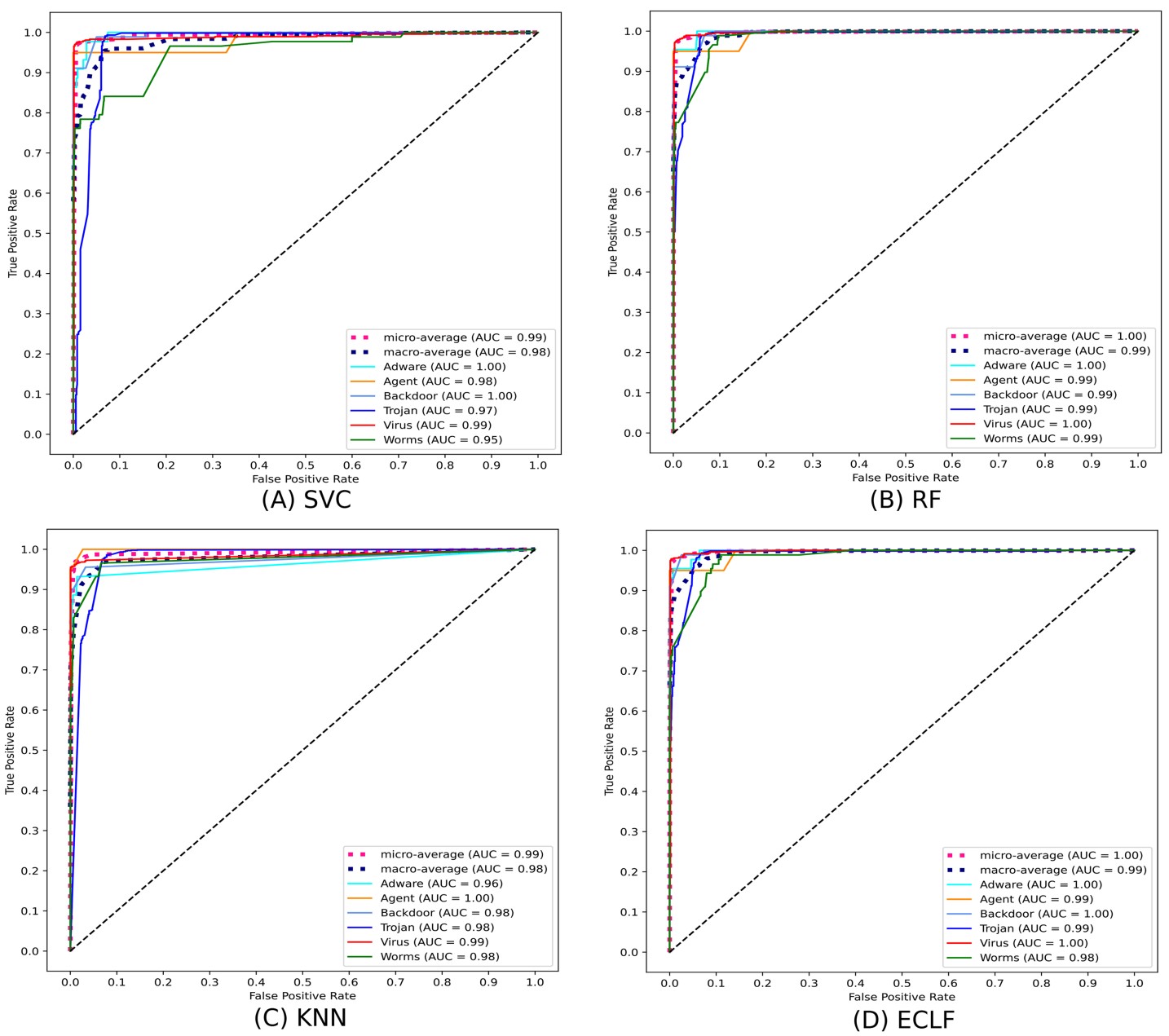

**Figure 14 (A–D) ROC-AUC of models for the iteration with maximal macro F1 of VirusSample.**

and 14 for VirusSample and Figs. 15 and 16 for MAL-API-2019 show the confusion matrix and ROC curve with AUC scores for all the models of the fold with maximum macro F1-score. Considering the principal diagonal of all three classifiers' confusion matrix, their classification ability is well justified. For all three datasets, the AUC scores of the ROC curve indicate the prominence in performance for multiclass malware classification problems on all the models using AMMC with AFS.

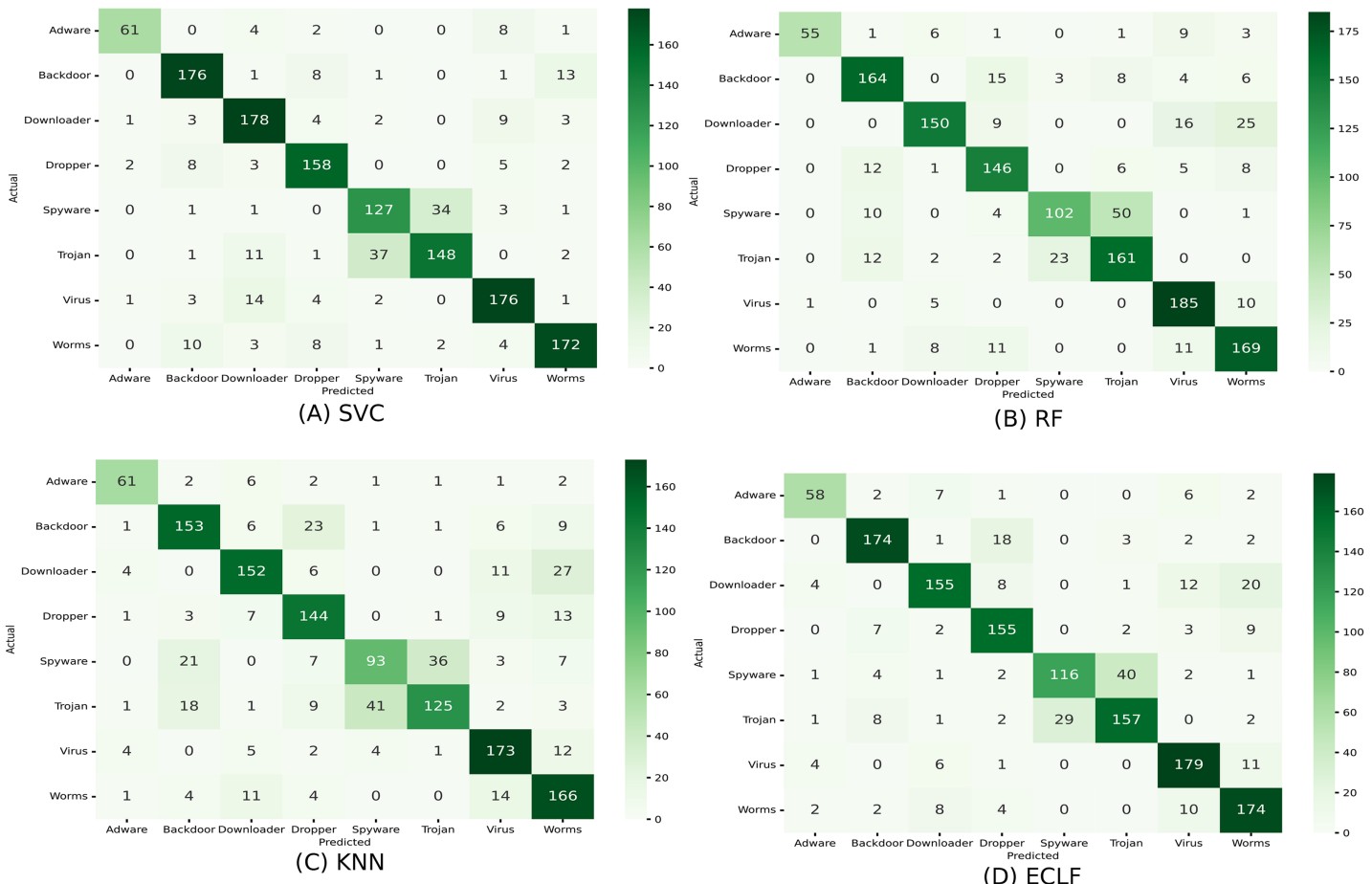

**Figure 15 (A–D) Confusion matrix of models for the iteration with maximal macro F1 of MAL-API-2019.**

## Case study

During the experiment, the case study on different malware classes of the three benchmark datasets revealed notable findings on their possible detection rate. As all the datasets are imbalanced, the precision-recall curves are plotted to emphasize all the classifiers' positive class prediction performance. The identification of Spyware and Trojans is more complicated than that of the other six malware classes: Adware, Backdoor, Downloader, Dropper, Virus, and Worms in the dynamic API sequence-based unbalanced dataset MAL-API-2019 even though they have a good number of samples. However, even with a smaller sample size for Adware, all the classifiers showed a better detection rate than Spyware and Trojan. The precision-recall curves for the classifiers SVC, RF, KNN, and ECLF are plotted in Fig. 17, and all the classifiers agree with lower precision-recall AUC scores for Trojan and spyware in contrast to other classes.

Identifying the class Worms is more complicated than all other classes in the static API sequence-based imbalanced datasets VirusShare and VirusSample. The detection rates of

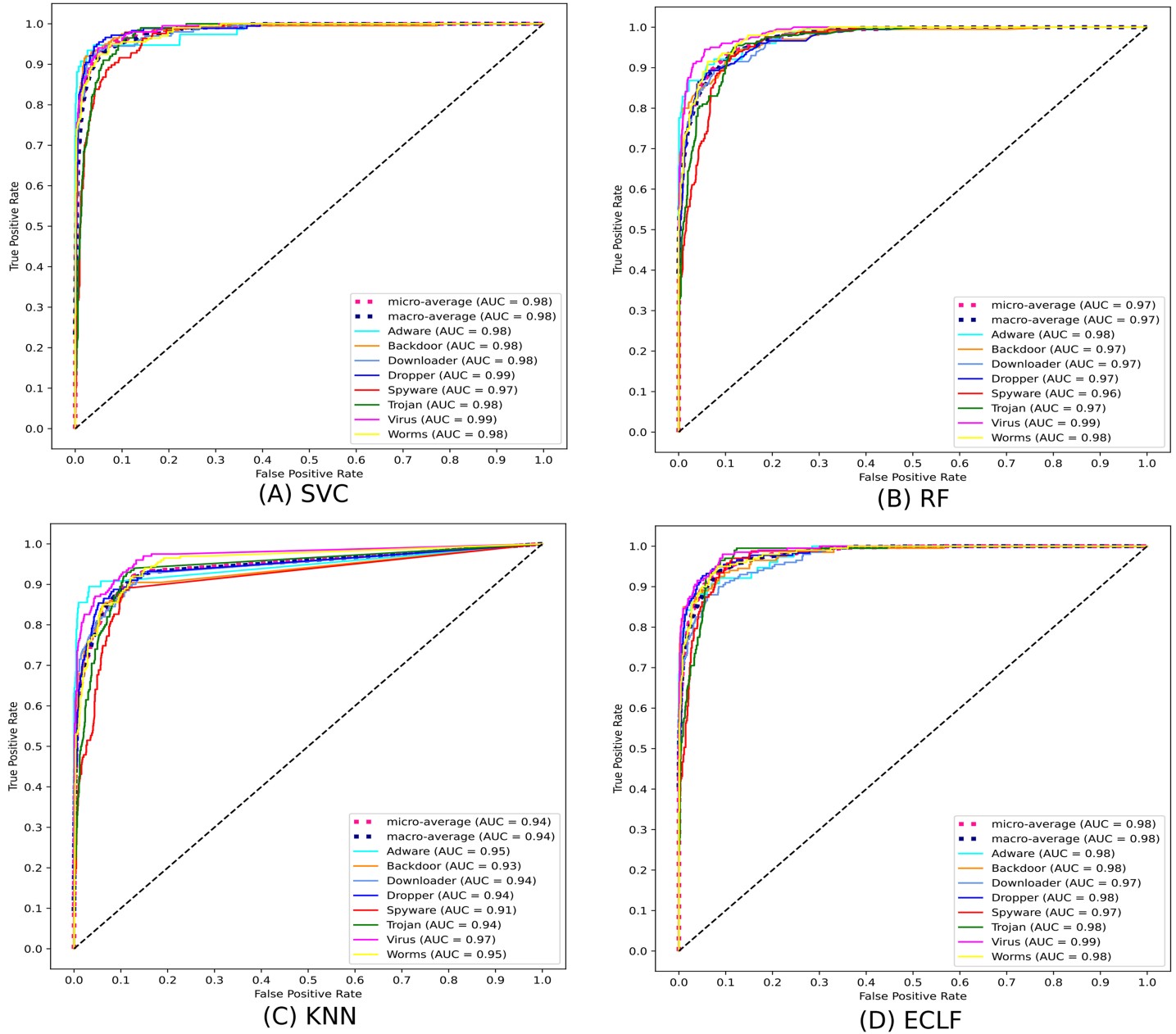

**Figure 16** (A–D) ROC-AUC of models for the iteration with maximal macro F1 of MAL-API-2019.

Adware, Agent, and Backdoor are found to be relatively higher than that of Worms with the VirusSample dataset, even though their sample sizes are nearly equal. In the VirusShare dataset, the detection rate of Worms and Agent is found to be lesser than that of all other classes. For both VirusShare and VirusSample, due to the presence of a higher number of samples for Virus and Trojan, the detection rate is very high compared to others in all the classifiers. The precision-recall curves plotted in Figs. 18 and 19 also show lower precision-

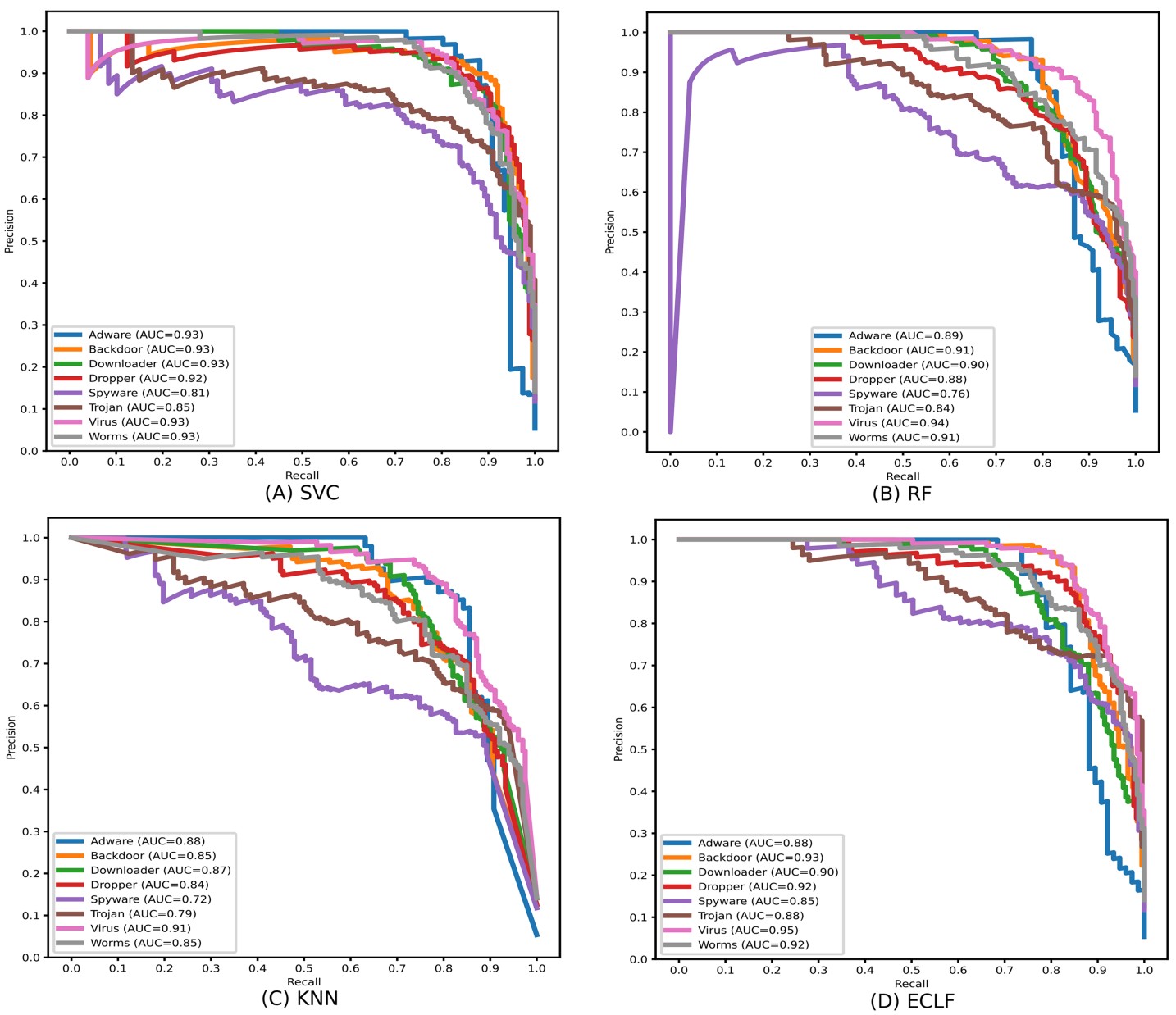

**Figure 17 (A–D) Precision-recall curves of models for the iteration with macro F1 with MAL-API-2019.**

recall AUC scores for the class Worms on all the classifiers with VirusShare and VirusSample datasets, respectively.

## Comparison with other works

In this article, the framework is experimented on the aforementioned three open benchmark datasets. Many researchers have considered these datasets to evaluate their model. Their findings helped to make a convincing comparison. Tables 10 and 11

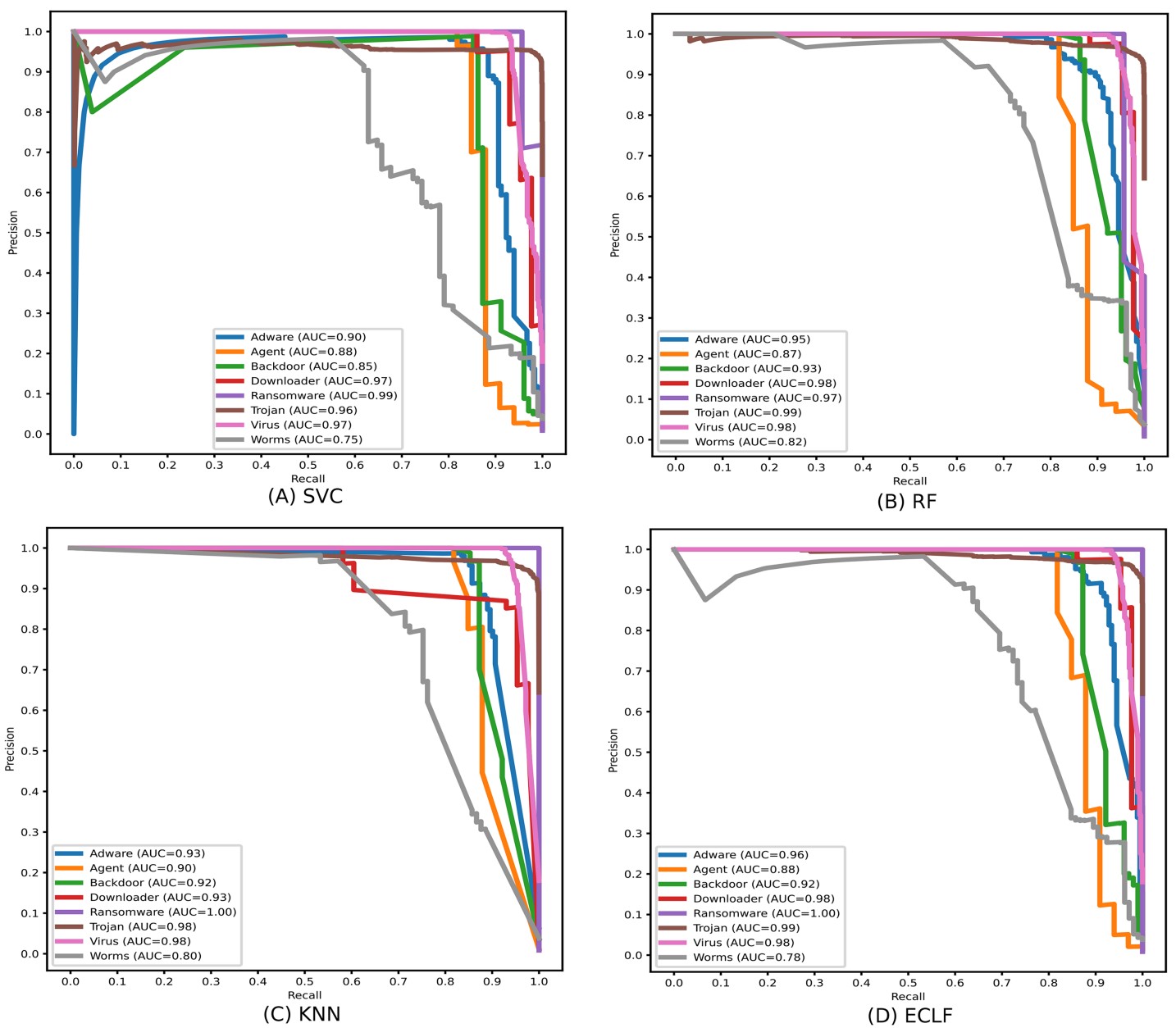

**Figure 18 (A–D) Precision-recall curves of models for the iteration with maximal macro F1 with VirusShare.**

compares earlier researchers' results against AMMC with AFS for VirusShare and VirusSample datasets. *Düzgün et al. (2021)* have worked on an imbalanced version of both datasets, where the maximum macro F1-score is 0.73 using SVM on VirusSahre and 0.78 using LSTM on VirusSample. They have also worked on a balanced version of both datasets where the maximum macro F1-score is 0.76 using SVM on VirusShare and 0.91 using CANINE on VirusSample. *Demirkiran et al. (2022)* got macro F1-scores of 0.72 and 0.80 using RTF on the imbalanced version of VirusSahre and VirusSample, respectively. Considering the above-mentioned best two results on the imbalanced version, our

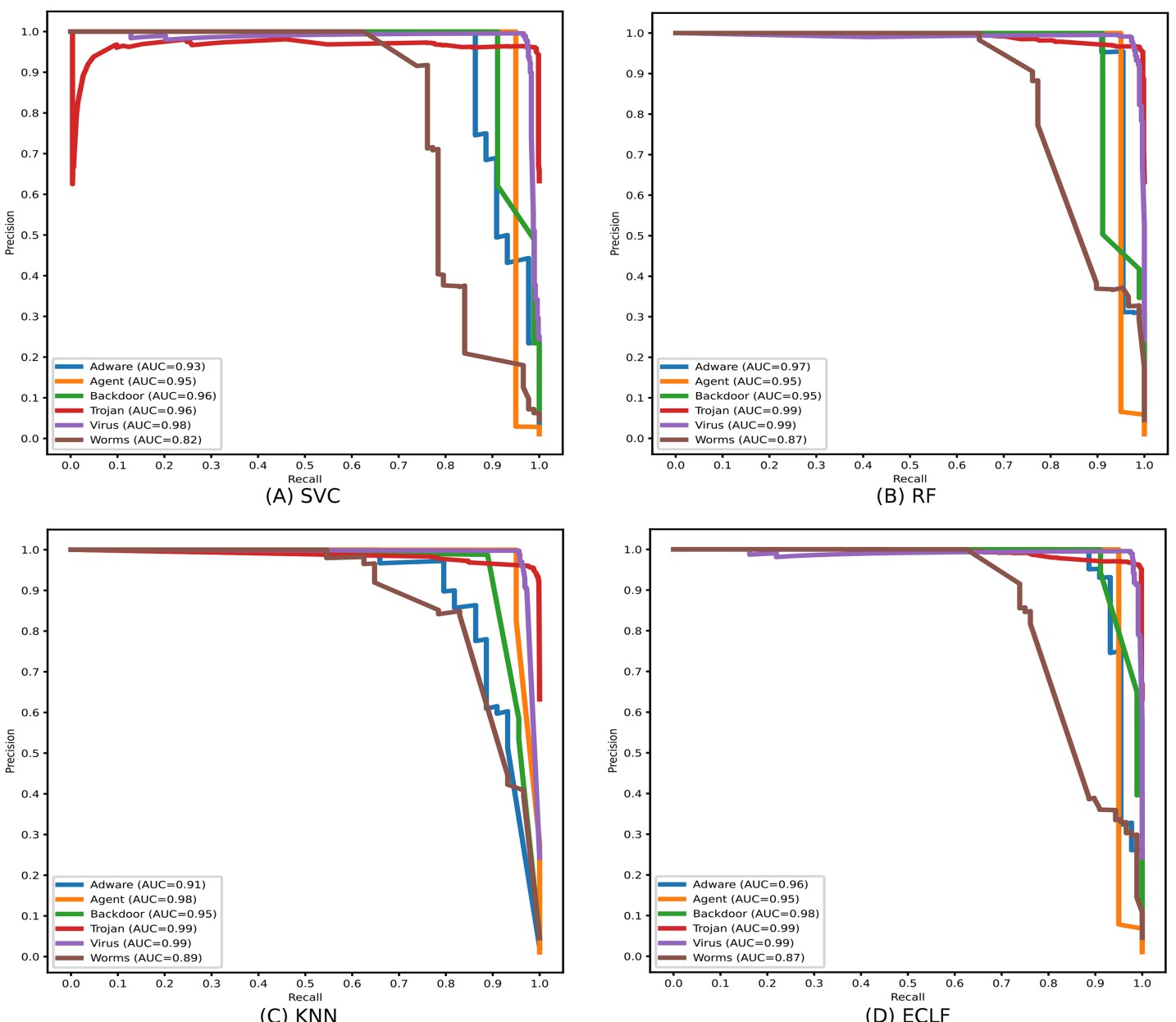

**Figure 19  (A–D) Precision-recall curves of models for the iteration with maximal macro F1 with VirusSample.**

**Table 10  Comparison on VirusShare.** Bold values indicate the best performance.

|  | Models | Macro F1 | ROC-AUC |
|---|---|---|---|
| *Düzgün et al. (2021)* | SVM | 0.73 | 0.92 |
|  | XGBoost | 0.72 | 0.96 |
| *Düzgün et al. (2021)* (balanced) | SVM | 0.76 | 0.94 |
|  | XGBoost | 0.75 | 0.96 |

(*Continued*)

| | Models | Macro F1 | ROC-AUC |
|---|---|---|---|
| *Demirkiran et al. (2022)* | RTF | 0.72 | 0.95 |
| *Miao et al. (2024)* | DistillMal | 0.69 | – |
| | BERT-base | 0.70 | – |
| **AMMC with AFS (the proposed framework)** | **RF** | **0.92** | **0.99** |

**Table 11  Comparison on VirusSample.** Bold values indicate the best performance.

| | Models | Macro F1 | ROC-AUC |
|---|---|---|---|
| *Düzgün et al. (2021)* | XGBoost | 0.74 | 0.98 |
| | LSTM | 0.78 | 0.97 |
| | CANINE | 0.72 | 0.96 |
| *Düzgün et al. (2021)* (balanced) | XGBoost | 0.90 | 0.99 |
| | LSTM | 0.84 | 0.95 |
| | CANINE | 0.91 | 0.98 |
| *Demirkiran et al. (2022)* | RTF | 0.80 | 0.97 |
| *Miao et al. (2024)* | DistillMal | 0.60 | – |
| | BERT-base | 0.56 | – |
| **AMMC with AFS (the proposed framework)** | **RF** | **0.94** | **0.99** |

**Table 12  Comparison on MAL-API-2019.** Bold values indicate the best performance.

| | Models | Macro F1 | ROC-AUC |
|---|---|---|---|
| *Ferhat Ozgur et al. (2020)* | Single Layer LSTM | 0.47 | – |
| *Li & Zheng (2021)* | GRU (Case-2) | 0.57 | – |
| *Demirkiran et al. (2022)* | RTF | 0.61 | 0.88 |
| *Cannarile et al. (2022)* | ExtraTree | 0.58 | 0.75 |
| *Avci, Tekinerdogan & Catal (2023)* | CNN LSTM | 0.24 | 0.83 |
| *Galli et al. (2024)* | BiLSTM | 0.54 | – |
| *Qian & Cong (2024)* | CAFTrans | 0.65 | 0.89 |
| **AMMC with AFS (the proposed framework)** | **SVC** | **0.84** | **0.98** |

framework AMMC with AFS outperformed all for both datasets with macro F1-scores of 0.92 and 0.94, respectively. Similarly, Table 12 highlights the superior performance of AMMC on MAL-API-2019 with a macro F1-score of 0.84 using SVC compared to the macro F1-scores of 0.65, 0.61, 0.58, 0.57, 0.54, and 0.47 achieved by others.

## LIMITATIONS AND FUTURE WORK

In this work, the iterative feature selection technique AFS goes through many iterations to find a better feature weight. Hence, the additional time of AFS may be an overhead in

terms of the total model training time. AFS employs a greedy technique to search weights to select influential APIs in an API sequence. AFS terminates after a fixed number of iterations when there is no evidence of performance improvement. Deciding on the terminating iteration depends on the dataset.

Future work will be devoted to exploring the usefulness of other optimization techniques in contrast to the greedy approach in AFS for a better classification score. The selection of optimal weights through AFS from one of the best-performing base machine learning models can be applied to deep learning techniques. It may highlight better classification abilities in the case of multiclass malware classification problems. A recursive feature selection technique to select influential APIs from API sequences for multiclass malware datasets may be contrasted with the proposed iterative feature selection technique AFS.

# CONCLUSION

This research presents an AMMC framework on variable length API sequences for efficient imbalanced multiclass malware classification. The primary goal has been to develop a technique for adaptively selecting features (*i.e.*, APIs) by class using AFS and demonstrate its applicability and usefulness in multiclass malware classification situations. The greedy search approach used by AFS is fundamental to this work. The described studies on three open multiclass malware datasets, VirusSahre, VirusSample, and MAL-API-2019, demonstrated, as stated in the findings section, the efficacy of AMMC employing AFS. AMMC ensures macro F1 values of 0.92, 0.94, and 0.84 for VirusSahre, VirusSample, and MAL-API-2019, with macro ROC-AUC scores of 0.99, 0.99 and 0.98, respectively. The comparison of AMMC's performance with the results of other authors on the same datasets reveals considerable improvements. Furthermore, the results of all of the base machine learning models and the ensemble model outlined in Tables 7–9 in this research performed better with AMMC than the results of others shown in Tables 10–12. Exploring other strategies to ensure a better selection of APIs in varying length API sequences on API sequence-based malware datasets will increase detection efficiency even more.

## Funding

The authors received no funding for this work.

## Competing Interests

The authors declare that they have no competing interests.

## Author Contributions

- Binayak Panda conceived and designed the experiments, performed the experiments, analyzed the data, performed the computation work, prepared figures and/or tables, authored or reviewed drafts of the article, and approved the final draft.

- Sudhanshu Shekhar Bisoyi conceived and designed the experiments, performed the experiments, analyzed the data, performed the computation work, prepared figures and/ or tables, authored or reviewed drafts of the article, and approved the final draft.
- Sidhanta Panigrahy conceived and designed the experiments, performed the experiments, analyzed the data, performed the computation work, prepared figures and/ or tables, authored or reviewed drafts of the article, and approved the final draft.
- Prithviraj Mohanty conceived and designed the experiments, analyzed the data, prepared figures and/or tables, and approved the final draft.

## Data Availability

The MAL-API-2019 Dataset is available at Mendeley: Çatak, Ferhat Özgür (2019), "Mal-API-2019", Mendeley Data, V2, DOI 10.17632/w393cchcb7.2.

The VirusShare and VirusSample Datasets are available at GitHub: https://github.com/ khas-ccip/api_sequences_malware_datasets.

The source codes are available in the Supplemental File.

## Supplemental Information

Supplemental information for this article can be found online at http://dx.doi.org/10.7717/ peerj-cs.2752#supplemental-information.

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
