# Peer review of "Machine learning techniques for imbalanced multiclass malware classification through adaptive feature selection"

_PeerJ Computer Science, doi:10.7717/peerj-cs.2752_

## Round 0.1 · original submission · Major Revisions

Based on the referee reports, I recommend a major manuscript revision. The author should improve the manuscript, carefully consider the reviewers' comments in the reports, and resubmit the paper.

Reviewer 1 ·

Basic reporting

Authors should carefully address each of the following comments and concerns, giving thoughtful consideration to any suggested improvements. If certain comments cannot be fully addressed, these should be acknowledged as limitations within the study’s limitations section. Clearly identifying these areas will enhance the transparency and rigor of the research.



Abstract:

Malware programs evolve as technology advances, threatening our security and privacy. As the number of malware variants proliferates, conventional signature-based malware detection has become less effective and time-consuming. Artificial intelligence techniques, such as Machine Learning and Deep Learning, are becoming increasingly popular for studying and analyzing malware behavior in API call sequences to detect it more efficiently. Machine learning models use fewer resources to train than deep learning models.
* This background is essential and should be provided in the introduction of the paper, as it sets the stage for the proposed approach and highlights the need for more efficient and scalable malware detection methods.
* Focus on Multiclass Problem: The abstract should emphasize that this is a multiclass classification problem, specifying the number of classes and the best classification results obtained.
* Clarity of Results: Instead of using vague terms like "promising results," the abstract should provide specific details about the best classification performance, including relevant metrics for the datasets.
* Methodology Summary: The abstract should succinctly describe the key methodology without overwhelming the reader with excessive details, focusing on the core components such as Skip-gram embeddings and AFS.
* This revised abstract would provide a clearer, more focused overview of the study, emphasizing the multiclass classification aspect and the results achieved.

* The labels in your graphs should be clearer and more readable to ensure that the data is communicated effectively. Increasing the label size, choosing appropriate font and color combinations, ensuring labels do not overlap with the background, and providing adequate spacing can improve readability. Additionally, keeping axis titles and data series descriptions concise and clear will help viewers understand the graphs more easily.

The background of the study and the review of related literature are sufficiently thorough and provide a solid foundation for the research. The study effectively builds upon existing work in the field, clearly highlighting the limitations of conventional malware detection methods and the growing importance of artificial intelligence techniques. Relevant and recent studies are well-integrated, showcasing the advancements in machine learning and deep learning applications for malware analysis.

Experimental design

*In the methodology section, the three datasets used should be clearly explained, including the variables within each dataset and the class distributions.

Data Preprocessing Methods:
The methodology lacks a detailed explanation of data preprocessing steps such as feature scaling, normalization, or handling missing data. These steps are essential for improving model performance, especially for algorithms like SVM or KNN.

Model Evaluation Metrics:
The methodology focuses on accuracy but does not consider other important metrics such as ROC-AUC, which would provide a more comprehensive assessment of model performance, especially in imbalanced datasets.

Feature Selection:
While AFS is used for feature selection, the methodology does not explain how to handle multicollinearity or how to validate the chosen features' relevance. Additional methods such as recursive feature elimination or regularization could be considered.

Model Comparison:
The methodology mentions SVM and Neural Networks but does not justify why these models were selected or how they compare to other possible algorithms like decision trees, random forests, or gradient boosting.

Hyperparameter Tuning:
The methodology does not include any mention of hyperparameter tuning, which can significantly impact the performance of the models. Implementing techniques like grid search or random search should be considered to optimize the models.

Validity of the findings

The results of this study are both sufficient and provide a comprehensive contribution to the field. The findings demonstrate the effectiveness of the proposed approach in addressing the challenges of multiclass malware classification, especially in the context of imbalanced datasets. The promising performance achieved across multiple datasets validates the robustness and applicability of the model. Additionally, the use of novel techniques like Adaptive Feature Selection and Skip-gram API embeddings presents a valuable advancement in malware detection methods.

Additional comments

Authors should carefully address each of the following comments and concerns, giving thoughtful consideration to any suggested improvements. If certain comments cannot be fully addressed, these should be acknowledged as limitations within the study’s limitations section. Clearly identifying these areas will enhance the transparency and rigor of the research.

Additionally, including a "Validity and Limitations" section under the discussion would enhance the clarity of the work by summarizing its accuracy and outlining its limitations.

Reviewer 2 ·

Basic reporting

This paper proposes a novel framework called Adaptive Multiclass Malware Classifier (AMMC) to address the issue of imbalanced multiclass malware classification. AMMC leverages a Adaptive Feature Selection (AFS) technique to train machine learning models using Skip-gram API embeddings. The paper applies the framework to three imbalanced malware datasets: VirusShare, VirusSample, and MAL-API-2019. Experimental results suggest that AMMC significantly improves performance metrics such as accuracy, precision, recall, and F1-score for malware classification. The paper is promising but still there are some questions which need to answer by authors:
Some sections, such as "Methodology," are overly verbose and could benefit from improved conciseness. For instance, the description of Algorithm 1 (AFS) should focus on essential details and avoid redundant explanations.
Although relevant, some figures (e.g., Figure 3) lack sufficient annotation to make them self-explanatory. Ensure that all visualizations and tables are fully labeled and described in the text.
While the paper is generally well-written, there are instances where the language is unclear. For example, in describing the novelty of AFS, certain phrases (e.g., "preserves the class imbalance problem") are ambiguous and need refinement.

Experimental design

While the paper compares AMMC to traditional machine learning models (e.g., SVC, RF, KNN), it lacks a discussion of more recent state-of-the-art methods for malware classification. Including comparisons to deep learning models like CNNs or LSTMs would strengthen the evaluation (cite this paper: Unveiling vulnerabilities in deep learning-based malware detection: Differential privacy driven adversarial attacks).

Validity of the findings

The paper reports performance metrics but does not provide statistical tests (e.g., t-tests or ANOVA) to validate the significance of the improvements.
The paper does not discuss the limitations of the proposed framework. For instance, the scalability of AMMC to larger datasets and its computational overhead should be evaluated.

Additional comments

Related work section need to improve by adding important methods in feature selection ( cite this paper: A new approach for feature selection in intrusion detection system)

Reviewer 3 ·

Basic reporting

no comment

Experimental design

Research gap has to be addressed. Methodology section should be in detail and it should be readable and understandable to the naive readers.

Validity of the findings

Results should be compare with other recent state-of-the-art works. some ablation study has to be conducted.

Additional comments

Review Comments:
1. Authors should define all the objectives of the work in a clearer way.
2. Authors should include some more recent works in Related work section.
3. Authors are suggested to include a table with summary of related works and remarks.
4. Addressing some research gap is required at the end of related work section.
5. Dataset details with description should be included.
6. Authors are suggested to conduct some ablation study to evaluate the proposed work.
7. Suggested to include experimentation section with some case studies.
8. Authors should compare their results with other recent works.

---

## Round 0.2 · accepted · Accept

The author has addressed the reviewer's comments properly, so I think the manuscript can be published.

Reviewer 1 ·

Basic reporting

The suggested comments regarding basic reporting have been addressed in detail and implemented by the authors.

Experimental design

With the latest revisions, the experimental design and methodology of the study adequately reflect the purpose and workflow of the article.

Validity of the findings

Comparison, verification and referencing of the findings of the study are sufficient for the scope of this article.

Additional comments

Dear Authors,

I have reviewed your paper titled "Machine Learning Techniques for Imbalanced Multiclass Malware Classification through Adaptive Feature Selection" in two stages and provided numerous comments and suggestions throughout the process.

I believe that in its final revised form, your paper will make a significant contribution to the field by proposing an adaptive feature selection method to address the imbalanced multiclass malware classification problem.

The critiques and suggestions I shared during the review process were intended to strengthen your work. I am pleased to observe that, through your comprehensive revisions and meticulous efforts, most of my concerns have been addressed. In particular, the detailed elaboration of your methodology, clearer presentation of experimental results, and explicit discussion of the limitations of your work have significantly enhanced the scientific quality of your paper.

I would like to thank you for your dedication and scientific rigor in addressing the shortcomings of the study. I am confident that your work will serve as a valuable reference for researchers in the fields of malware analysis and machine learning.

Reviewer 2 ·

Basic reporting

The authors have answered to my questions.

Experimental design

N/A

Validity of the findings

N/A

Additional comments

N/A